

# W-band Radar Observations for Fog Forecast Improvement: an Analysis of Model and Forward Operator Errors

Alistair Bell[a], Pauline Martinet[a], Olivier Caumont[a], Benoît Vié[a], Julien Delanoë[b], Jean-Charles Dupont[c], and Mary Borderies[a]

[a]CNRM, Université de Toulouse, Météo-France, CNRS, Toulouse, France
[b]Laboratoire Atmosphères, Milieux, Observations Spatiales/UVSQ/CNRS/UPMC, Guyancourt, France
[c]Institut Pierre Simon Laplace (IPSL), École Polytechnique, UVSQ, Université Paris-Saclay, 91128 Palaiseau Cedex, France

**Correspondence:** Alistair Bell (alistair.bell@meteo.fr)

**Abstract.** The development of ground based cloud radars offers a new capability to continuously monitor the fog structure. Retrievals of fog microphysics is key for future process studies, data assimilation or model evaluation, and can be performed using a variational method. Both the one-dimensional variational retrieval method (1D-Var) or direct 3D/4D-Var data assimilation techniques rely on the combination of cloud radar measurements and a

background profile weighted by their corresponding uncertainties to obtain the optimal solution for the atmospheric state. In order to prepare the exploitation of ground-based cloud radar measurements for future applications based on variational approaches, the different sources of uncertainty due to instrumental errors, background errors and the forward operator used to simulate the radar reflectivity need to be properly treated and accounted for. This paper aims at preparing 1D-Var retrievals by analysing the errors associated with a background profile and a forward

operator during fog conditions. For this, the background was provided by a high-resolution numerical weather prediction model and the forward operator by a radar simulator. Firstly, an instrumental dataset was taken from the SIRTA observatory near Paris, France for winter 2018-19 during which 31 fog events were observed. Statistics were calculated comparing cloud radar observations to those simulated. It was found that the accuracy of simulations could be drastically improved by correcting for significant spatio-temporal background errors. This was achieved

by implementing a most resembling profile method in which an optimal model background profile is selected from a domain and time window around the observation location and time. After selecting the best background profile a good agreement was found between observations and simulations. Moreover observation minus simulation errors were found to satisfy the conditions needed for future 1D-var retrievals (un-biased and normally distributed).





## 1 Introduction

The presence of fog is an issue for many modes of transport due to its effect of reducing visibility. When seen at airports, it can mean the grounding of flights, resulting in large economic costs due to delays and cancellations (Gultepe et al., 2007). Reliable fog forecasts, however, can allow for the planning of flights around a fog event, mitigating the impact it has. The development of high-resolution numerical weather prediction (NWP) models, with

horizontal resolutions in the order of 1 km, and vertical resolutions in the order of 10 m near the surface, offers the possibility to represent fog events with fine spatial and temporal resolutions. However, fog events are generally still poorly forecast with current NWP models (Steeneveld et al., 2015; Philip et al., 2016).

Fog is defined as the reduction of visibility below 1 km at the surface due to the presence of cloud droplets (Glickman and Zenk, 2000), and is thus strictly a boundary layer phenomenon. The lack of accurate observations inside the

boundary layer has in recent years become an increasingly discussed subject (NRC, 2009), and might contribute to the sub-optimal performance of high-resolution NWP models when forecasting boundary layer events, such as fog. Although traditional observation methods, such as radio soundings and in-situ surface observations provide the most accurate information, the development of ground-based remote sensing instruments offers measurements with a temporal resolution unmatched by traditional instruments. Thanks to these emerging technologies, new products

have been designed making use of observations from lidars, ceilometers and visibility meters to aid fog nowcasting, giving fog alerts with an average of 10 minutes to 50 minutes before fog formation (Haeffelin et al., 2016).

Recent developments in 95 GHz cloud radars have made these instruments much more affordable (Delanoë et al., 2016) allowing for cloud studies, including those on fog processes, to be performed with increased insight (Thies et al., 2010; Dupont et al., 2012; Wærsted et al., 2017). These have highlighted which physical processes are the

most important to improve in new models if fog characteristics are to be better represented. The assimilation of cloud radar data into an operational NWP model to give better fog forecasts with longer advance times, however, is yet to be seen.

A simple method for assimilating new observations into an NWP model is to first retrieve an atmospheric profile of a variable or set of variables, and to then assimilate this retrieved profile. Retrievals can be made through different

methods (statistical laws, optimal estimations (OE) (Maahn et al., 2020) using so-called one dimensional variational (1D-Var) retrievals of state variables (Martinet et al., 2015)). This study focuses on the preparation of future OE using 1D-Var data assimilation methods such as was performed in the work of Martinet et al. (2015, 2017) for temperature and humidity profiles.

These retrievals may then be used in a second step with a three/four dimensional variational data assimilation

(3D/4D-Var) scheme (Bauer et al., 2006; Janisková, 2015), or as a preliminary step towards direct variational data assimilation of the cloud radar reflectivity (Fielding and Janiskova, 2020). In order to first perform the 1D-Var retrieval, observations should be combined with an 'a priori' profile, otherwise known as a 'background' profile. Though this may be taken from climatological data, the more accurate the background profile, the more accurate





the final retrieval is likely to be (Rodgers, 2000). As commonly used in data assimilation, the background profile
considered in this study comes from a high-resolution NWP model—in this case the French convective-scale model
AROME (Seity et al., 2011), valid at the time and location of the retrieval.

The background profile must also be of the same variable type as the observation is made in. In the case of
remote sensing instruments this requires either a 'backward' model, to transform the observation variables into
those produced by the NWP model, or a 'forward' model to transform the variables given by an NWP model to
those made by the instrument. Due to the ill-posed nature of transforming radar reflectivity measurements into
LWC estimates (Atlas, 1954; Bohren and Huffman, 2008; Maier et al., 2012), the forward model approach has been
chosen in this study.

In order to make a 1D-Var retrieval, it is also necessary that the errors associated with the background and the
observations are properly modelled (Rodgers, 2000). For successful variational retrievals to be made, it is assumed
that i) the distribution of errors should follow a normal distribution and ii) that there should be no systematic bias in
the error distributions (Bouttier and Courtier, 2002). Background errors are due to inaccuracies in NWP forecasts.
The forward model may contain errors as a result of the hypotheses needed to simulate the observations, such as
assumptions on the cloud droplet size distribution in the context of radar reflectivity. Observations errors are due
to calibration uncertainties (Toledo et al., 2020; De Angelis et al., 2017), instrumental drifts and random noise.
The modelling of the errors associated with the background, the observations and the forward operator can be
difficult to specify for a given retrieval, owing to dependencies on the type of weather conditions observed, or the lead
time of the forecast used as a background profile, for example. However, an improved knowledge of background and
observation errors is required before the assimilation of any new observation type. The aim of this work is thus to
investigate the types of systematic and random errors which may be present in the three sources of errors previously
mentioned focusing on newly developed 95 GHz cloud radar during fog conditions.

This study has been performed using a dataset from the SIRTA observation site near Paris (Haeffelin et al.,
2005) which hosts a 95 GHz cloud radar, a ground-based microwave radiometer and other remote sensing and in-situ
instruments making continuous measurements. Up to 3h forecasts from the AROME model were used in conjunction
with a radar simulator, also referred to as observation operator or forward operator, designed for airborne 95 GHz
cloud radar (Borderies et al., 2018).

In this article, firstly an overview is given of the fog events used in this study. The performance of the AROME
model is then analysed by using a range of instruments to compare to the observed event. A method is then outlined
for the selection of a background profile which is expected to optimise future retrievals. Statistics are then presented
showing reflectivity innovations and the improvement gained through the profile selection method.



## 2 Dataset

### 2.1 SIRTA Observatory

All observations for this study were made at SIRTA (Site Instrumental de Recherche par Télédétection Atmosphérique) (Haeffelin et al. (2005)). Geographically, the site is located in the suburbs, about 20 km south of Paris, on the campus of the École polytechnique in Palaiseau, which is a semi-urban environment with trees, fields, houses and some industrial buildings. The observatory sits on a relatively flat plateau at around 160 m above sea level (asl). The period between 01/11/2018 and 19/02/2019 was analysed due to the relatively high concentration of fog events seen throughout this period.

### 2.2 Basta Cloud Radar

The cloud radar used in this study is a 95 GHz frequency-modulated continuous wave (FMCW) Doppler radar named the Bistatic Radar System for Atmospheric Sounding (BASTA, Delanoë et al., 2016). The instrument is a product of recent developments aimed at producing an inexpensive radar system to be used operationally. For this reason, the normally expensive high-powered pulsed transmitter has been replaced with a continuous transmitter with frequency modulation, to allow for the backscatter power and the line of sight velocity from the targets—in this case cloud droplets—to be determined. The benefit of using a cloud radar with a 95 GHz transmission frequency compared to radars using lower frequencies is in the sensitivity to cloud droplets. Where the Rayleigh approximation is valid, the power of the reflected radiation will be proportional to the sixth power of the radius of a spherical droplet and inversely proportional to the forth power of the wavelength of light incident. Thus, for a given transmitted power, radars operating at a higher frequency will have a greater sensitivity to smaller droplets. It does mean, however, that when large particles such as rain, hail or graupel are encountered, the signal can become quickly attenuated Kollias et al. (2007).

For monostatic radars, the receiver must be switched off during the transmission of a pulse, meaning that signal backscattered close to the radar cannot be detected, and a minimum detectable range of over 100m is typical for cloud radars sounding in a boundary layer mode (Liu et al. (2017)). The fact that BASTA employs a seperate receiver and transmitter (bistatic) thus allows the minimum measurement distance of the radar to be relatively small compared to that of a monostatic radar. It is capable of making measurements as close as 40 m above ground level, though the minimum detectable measurement values are quite high at this distance ($\approx -25$ dBZ for BASTA-SIRTA). This is due to the interaction between the antennas of the transmitter and receiver at close distances. The radar operates in 3 different modes with vertical resolutions ranging from 12.5 m to 100 m and maximal measurement distance from 12 km to 18 km respectively. For the BASTA-SIRTA, a three-second integration time is used, and the three different modes are cycled through continuously. This therefore gives observations for each mode once every 9 seconds.

The uncertainty associated with BASTA measurements will vary with usage and meteorological conditions. From a comparison with radar reflectivity simulations with rain rates over $2\,\mathrm{mm\,h^{-1}}$ the estimated uncertainty, providing





**Table 1.** Instruments used at SIRTA observatory

| Instrument Name | Measured Variable | Units | Measurement Uncertainty | Measurement Range |
|---|---|---|---|---|
| CL31 Ceilometer | Cloud Base Height | m | Greater of $1\%$ or $\pm 5\,\mathrm{m}$ | $7.5\,\mathrm{m}$ to $7.5 \times 10^3\,\mathrm{m}$ |
| CMP22 | Global Shortwave downwelling | $\mathrm{W\,m^{-2}}$ | $\pm 5\,\mathrm{W\,m^{-2}}$ | $0\,\mathrm{W\,m^{-2}}$ to $4000\,\mathrm{W\,m^{-2}}$ |
| DF-320 Visibility Sensor | Meteorological Optical Range | m | $10\%$ (up to $5\,\mathrm{km}$) | $0\,\mathrm{m}$ to $70 \times 10^3\,\mathrm{m}$ |
| Guilcor PT100 | 2 m Temperature | °C | $\pm 0.15\,°\mathrm{C}$ (at $0\,°\mathrm{C}$) | $-200\,°\mathrm{C}$ to $700\,°\mathrm{C}$ |
| HATPRO Microwave Radiometer | Liquid Water Path | $\mathrm{g\,m^{-2}}$ | $\pm 20\,\mathrm{g\,m^{-2}}$ | $0\,\mathrm{km}$ to $10\,\mathrm{km}$ |
| PM Rain Gauge 3030 | Precipitation Rate | $\mathrm{mm\,min^{-1}}$ | $\pm 8\%$ | $0\,\mathrm{mm\,h^{-1}}$ to $240\,\mathrm{mm\,h^{-1}}$ |
| Vector A100R anemometer | Wind Speed | $\mathrm{m\,s^{-1}}$ | $\pm 0.1\,\mathrm{m\,s^{-1}}$ ($< 10\,\mathrm{m\,s^{-1}}$) | $0.2\,\mathrm{m\,s^{-1}}$ to $70\,\mathrm{m\,s^{-1}}$ |

that the radome is not wet, is between $0.5\,\mathrm{dB}$ to $2.0\,\mathrm{dB}$ (Delanoë et al., 2016). A wet radome can affect readings by up to 14 dB. Below 230 m, the far field approximation, which is used to give the radar reflectivity value, is not valid.
An overlap correction, derived using rain events is therefore used to correct for this effect (Delanoë et al., 2016).

### 2.3 Other Instruments

In order to define fog events, the visibility at or near to surface height must be known. Though there has been work done to classify the visibility from radar reflectivity (Li, 2015), which was done with a Plan Position Indicator (PPI) scanning strategy, the lowest gates still suffered from quality issues due to ground clutter. The most reliable way
to measure the visibility is with a visibility metre. The visibility metre deployed at ground level at SIRTA is the Degreane Horizon DF320 visibility monitor. This is able to give the meteorological optical range from $5\,\mathrm{m}$ to $70\,\mathrm{km}$, with a measurement error under $5\,\mathrm{km}$ of $10\%$.

Ground-based microwave radiometers also provide insight into the fog properties through liquid water path retrievals. The HATPRO microwave radiometer (Rose et al., 2005) operates in two spectral bands ($22\,\mathrm{GHz}$ to $31\,\mathrm{GHz}$
and $51\,\mathrm{GHz}$ to $58\,\mathrm{GHz}$) in order to make retrievals of the temperature and humidity profiles, integrated liquid water and water vapour contents providing information about the atmospheric stability. For this study, only the liquid water path retrievals were used. These retrievals have an expected accuracy of $20\,\mathrm{g\,m^{-2}}$ (Crewell and Löhnert, 2003).

A ceilometer was used primarily for the classification of fog types. Low cloud whose base is descending is very likely to be observed before an instance of cloud base lowering (CBL) fog. A Vaisala CL-31 ceilometer (Martucci
et al., 2010) was used to measure the cloud base height. This uses a pulse lidar to sense the cloud base and is capable of sensing up to three layers simultaneously with a range from $0\,\mathrm{km}$ to $7.6\,\mathrm{km}$.

The wind speed, temperature and rain rate at surface are also important parameters to sense when determining the fog events and classifying them. The specifications for the instruments used in this study are noted in table 1.



**Table 2.** Parameterisation schemes in AROME model

| Process | Scheme | Reference |
|---|---|---|
| Cloud microphysics | ICE-3 | Pinty and Jabouille (1998) |
| Long wave radiation | RRTM | Mlawer et al. (1997) |
| Short wave radiation | Computations of Solar Heating | Fouquart et al. (1980) |
| Surface fluxes | SURFEX | Masson et al. (2012) |
| Turbulence | Turbulence Scheme for Mesoscale and Large Eddy Simulations | Cuxart et al. (2000) |
| Urban features | TEB | Masson (2000) |

## 2.4 The AROME Model

The NWP model used in this study is the French convective-scale model AROME (Seity et al., 2011). AROME has been used operationally since 2008, but has since then seen improvements in the horizontal resolution, from 2.5 km to 1.3 km, and in the vertical resolution, which has advanced from 60 to 90 levels, with the first level starting 5 m above the surface. Near the surface, the vertical levels are aligned with the topography which are then spaced so as to follow isobars at the top of the model. The model covers a domain centred on France and encompassing most of 145 western Europe. A 3D-Var data assimilation cycle takes place once every hour.

The model was developed from the Meso-NH research model (Lafore et al., 1998; Lac et al., 2018), and therefore most of the model physics is resolved in the same way. A bulk one moment micro-physical scheme is used (ICE-3, Pinty and Jabouille, 1998) which fixes the droplet number concentration over land and sea and specifies six species of atmospheric water (graupel, ice, snow, rain, cloud liquid water over land and cloud liquid water over sea). An 150 analysis of the parameters used in ICE-3 and their effect on the distribution shape is given in section 4. Table 2 summarizes the parameterisation schemes relevant to fog processes with the corresponding references.

## 2.5 The Forward Operator

The forward operator used to convert the parameters supplied by the AROME model into radar reflectivity was developed by Borderies et al. (2018) and designed for vertically-pointing airborne W-band cloud radars. Input 155 variables include vertical profiles of pressure, temperature, humidity and the content of five hydrometeor types (rain, graupel, snow, ice and liquid cloud). From this, it simulates the reflectivity at the resolution of the input profiles with attenuation taken into account for hydrometeors and moist air. The Liebe (1985) model is used to calculate attenuation by moist air. The reflectivity calculations are consistent with the ICE-3 bulk microphysical scheme, which is operationally used in the AROME model. The sensitivity of the radar is also taken into account, 160 by limiting the minimum simulated reflectivity to the minimum observed reflectivity at each range gate.

Two versions of the radar simulator were developed: the one used in this work employs the Mie approximation (Wriedt, 2012) which models particles as spherical, and is a valid approximation for cloud liquid water droplets. A



**Table 3.** Number of fog types observed at the Sirta observation site between 01/11/2018 and 19/02/2019

| Cloud-base lowering | Precipitation | Radiative | Advection | Unknown | Total |
|---|---|---|---|---|---|
| 14 | 4 | 10 | 0 | 3 | 31 |

version using a T-matrix method is also available for simulating reflectivity from hydrometeors with a more complex shape.

## 3 Investigation into Background Errors During Fog Conditions

1D-Var retrievals can be highly sensitive to the background profile as demonstrated by Ebell et al. (2017) in the context of LWC retrievals from MWR and 35 GHz cloud radar synergy. Background profiles are commonly provided by short-term forecasts from NWP models which are prone to errors of different nature, such as temporal and spatial errors. This section aims at better understanding typical errors from the AROME background profiles during fog conditions.

### 3.1 Overview of the observed fog events

Fog can occur through several atmospheric processes, not all of which are modelled equally well. Philip et al. (2016) has shown that the AROME model seems to succeed in predicting certain types of fog better than others. Notably, CBL events are badly predicted compared to radiative fog. A simple fog classification based on the one described in Tardif and Rasmussen (2007) was performed on the instrumental dataset after updates in the suggested thresholds chosen in the classification. These updates concerned the precision of the conditions and reflected some misleading instrument readings. A total of 31 fog events were observed over the period, the numbers of each type are detailed in table 3. In line with previous studies performed by (Philip et al., 2016; Dupont et al., 2016) and which looked at fog events in Paris, and Román-Cascón et al. (2019), which examined fog events over a short period in January 2016 on the Spanish Northern Plateau, the majority of fog events were either cloud base lowering or radiative. Precipitation fog was the third most observed type, for which fog events were typically shorter than radiative or cloud base lowering. The quality of AROME short term forecasts during these 31 fog events is investigated in the next sections with a focus on spatial and temporal errors as well as typical fog parameters (duration, formation and dissipation times, thickness and liquid water content).

### 3.2 AROME forecast skill scores during fog conditions

In order to make a comparison between observed and modelled fog events, it is necessary to define an equivalent definition of fog events from parameters inside the AROME model. For this study, AROME forecasts were regenerated with outputs produced with a temporal period of 10 minutes, with lead times of 0 minutes to 180 minutes. The





forecasts were extracted for a $28\,\mathrm{km} \times 28\,\mathrm{km}$ domain centred on the SIRTA observatory site. Visibility in the AROME

model was diagnosed from a newly developed parameterisation based on the liquid water content profile according to Dombrowski-Etchevers et al. (2020), which has been used operationally to give a visibility output from the model since July 2019.

A comparison of the exact time match at which fog profiles were observed against when they were predicted was carried out. Visibility measurements were averaged over a $10\,\mathrm{min}$ period. Observed visibility values of lower than

$1\,\mathrm{km}$ were considered as fog. Observations where rain was sensed with the rain gauge and simulations in which rain was present in the bottom layer were not considered as fog. The accuracy of the model was then analysed by comparing each fog profile in the model against each fog profile from the averaged visibility. The commonly used contingency table based on this comparison is shown in table 4 where GD indicates cases of good fog detection, FA cases of false alarm, ND cases of missed fog events by the model and CN correct negatives.

Based on this table, the frequency bias index (FBI), which assesses the over- or under-prediction of an event, and critical success index (CSI) which assesses how well events are forecast, are calculated. These indices are defined in equations (1) and (2). The probability of detection (POD), the probability of an observed event being forecast, and false alarm ratio (FAR) the probability of a fog forecast being incorrect are also given ((3) and (4)).

$$\mathrm{FBI} = \frac{\mathrm{GD} + \mathrm{FA}}{\mathrm{GD} + \mathrm{ND}} \tag{1}$$

$$\mathrm{CSI} = \frac{\mathrm{GD}}{\mathrm{GD} + \mathrm{ND} + \mathrm{FA}} \tag{2}$$

$$\mathrm{POD} = \frac{\mathrm{GD}}{\mathrm{GD} + \mathrm{ND}} \tag{3}$$

$$\mathrm{FAR} = \frac{\mathrm{FA}}{\mathrm{GD} + \mathrm{FA}} \tag{4}$$

Scores of FBI and CSI were found to be 1.59 and 0.32 respectively. The scores agree well with the work of Philip et al. (2016) who calculated a score of 1.24 and 0.37 respectively as well as Martinet et al. (2020) who found scores

of 1.77 and 0.35. The FBI score indicates that the model over-predicts the occurrence of fog with a large number of false alarms and the CSI score means that only $32\,\%$ of fog events (observed and/or predicted) are correctly forecast by the model. The POD is $63\,\%$, meaning that background profiles of acceptable quality could be expected to be found at about this rate without any other selection method during fog events. With a $60\,\%$ FAR, this also highlights how large errors are made when the closest AROME grid point (both spatially and temporally) is used

during fog-clear scene. The next section investigates how much spatio-temporal variability affects fog forecast errors in the AROME model.





**Table 4.** Contingency table of fog profiles seen in the simulation and observations. Good detection (GD) occurs at the intersection of fog simulated and observed, false alarm (FA) where fog is simulated but not observed, unpredicted (ND) where fog is observed but not simulated, and correct negative (CN) where fog is neither predicted nor observed.

|  |  | Fog Simulated | | |
| --- | --- | --- | --- | --- |
|  |  | Yes | No | Total |
| **Fog Observed** | Yes | GD = 586 | ND = 349 | 935 |
|  | No | FA = 902 | CN = 13411 | 14313 |
|  | Total | 1488 | 13760 | 15248 |

## 3.3 Spatial and Temporal Error Analysis

Spatial and temporal errors refer to modelled fog events which are spatially and/or temporally displaced from the true event. These types of errors were examined to quantify how they can affect the forecast scores.

Firstly, spatial errors were examined by looking at the thickness of the fog layer over the 28 km × 28 km domain around the observation. Figure 1 shows an example of the development of a radiative fog event on 04/11/2018 which persisted for around eight hours in the model and around 5 hours in the observations. The surface height is shown in black contours on the figures, with the higher surfaces in the top left of the map. In the formation stage of the event, approximately half of the domain is covered by fog. The differences in fog thickness at this stage of the event

are around 100 m for the AROME grid points already covered by fog. At 05:00 UTC, in the mature phase of the event, the fog thicknesses have approximately the same variability as in the early formation stage, but almost all of the AROME grid points have fog conditions. It may also be noted that the thickest fog layers occur where surface height is the lowest showing how fog top heights are related to the topography— a subject that is beyond the scope of this work and has been widely discussed elsewhere (Müller et al., 2010; Ducongé et al., 2019). At 10:20 UTC,

shortly before the fog event ends, there is substantial variability of around 150 m, and in several AROME grid points the event has already dissipated. After 11:00 UTC, the fog layer lifts and disperses, and the modelled fog event ends throughout the whole domain.

     The significant variability in simulated fog thickness indicates that during the formation and dissipation phases of the fog event, increased value may be brought to the background accuracy by choosing a model profile which more

closely fits the observed atmospheric profile than the closest grid point.

     The temporal errors associated with fog forecasts were then examined. For each observed fog event, the corresponding starting and ending time in the model space was found by looking over a 12 hour window (±6 hours) around the observation. In the case that there were two events seen in the model within one observed event, the closest start and end times corresponding to the observations were taken. Out of 31 fog events observed, 21 could

be matched within the twelve hour window to a simulated event. The histograms in figure 2 show the distribution of hours for which fog was observed and simulated and the temporal differences in the formation time, dissipation

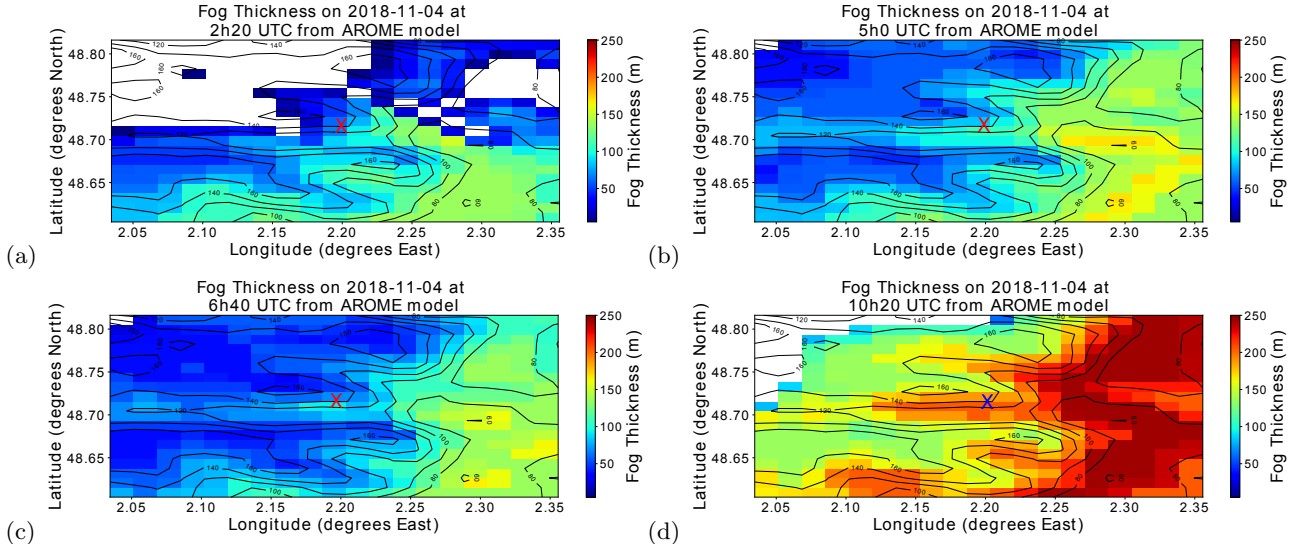

**Figure 1.** Fog top altitude above ground level in the AROME model during a radiative fog event. (a) in the formation phase of the event at 02:20 UTC, (b) in the mature phase at 05:00 UTC, (c) mature phase at 06:40 UTC, (d) in the dissipation phase at 11:00 UTC. The fog event ended at around 11:30 UTC in the model. The Sirta site is marked by the red or blue cross. Black contours represent the surface height.

time and duration of fog events observed. The diurnal cycle of fog events is generally well predicted by the model, with the majority of events taking place between midnight and late morning time. It may be seen with formation and dissipation time differences that most fog events which occur in both the observations and simulations have start and end time differences of less than three hours. The simulated events tend to form earlier (with a median of 25 minutes), and dissipate later (with a median of 20 minutes) than the observed events. Additionally, simulated fog events tend to have shorter duration, with an average fog time length of 4 hours 53 minutes (4H53M) compared to 6H03M for observed events, as many more shorter fog events were simulated than observed.

It was found that the rate of formation between 10:00 UTC and 20:00 UTC (not shown in figure2) was larger in the observations than in the model, whilst between 00:00 UTC and 8:00 UTC the model had a greater susceptibility to predict fog formation. This result indicates that the model over-predicts the rate of night fog and under-predicts the rate of afternoon fog, which could indicate that the radiation budget of the model could be improved.

### 3.4 Fog Property Error Analysis

In addition to spatial and temporal errors, the AROME background accuracy will depend on the capability of the AROME model to reproduce the vertical structure of fog microphysical properties. A radar-microwave radiometer combination enables the measurement of fog characteristics such as the layer thickness and the liquid water path of the fog layer. Analysis of a high resolution model's accuracy in predicting these variables has not been extensively





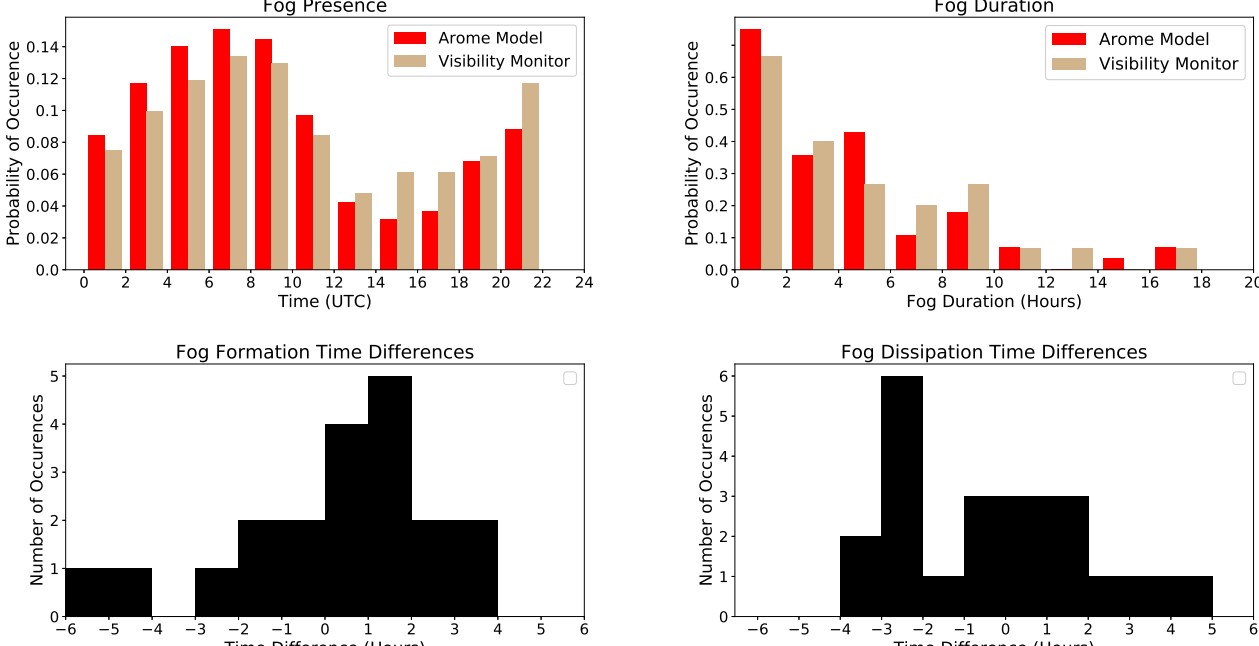

**Figure 2.** Top left to bottom right: Times at which fog was observed (brown bars) and simulated (red bars); duration of fog events observed and simulated; fog formation time differences for matching events; fog dissipation time differences for matching events (differences are positive where the event occurs later in the observation).

carried out in previous work, as without these instruments a labour intensive method involving tethered balloons or unmanned aerial vehicles (UAVs) is required. The fog layer thickness depends on the rate of cooling, the entrainment

and surface interactions among other processes. It was also demonstrated by Wærsted (2018) that the fog top height is a key parameter in determining the fog dissipation. It thus follows that the better the fog top height prediction, the better the fog dissipation forecast will be. This section aims at investigating fog thickness and LWP errors observed in the AROME fog forecasts during the winter 2018-2019.

Fog thicknesses were derived from the radar observations during fog conditions. This was found from the height at

which the radar reflectivity dropped below the larger of $-45\,\mathrm{dBZ}$ or the sensitivity of the radar at that range gate. The fog top height was then found in the model from the simulated reflectivity (with the same conditions) for times when fog conditions were simulated. The height resolution of the radar was $12.5\,\mathrm{m}$, whereas the resolution for the model ranged between $12\,\mathrm{m}$ at the surface to $65\,\mathrm{m}$ at $750\,\mathrm{m}$ agl, giving an uncertainty in fog top height difference of $12.25\,\mathrm{m}$ to $37.75\,\mathrm{m}$. Comparisons were made between the two, for times when both observations and simulations are

under fog conditions. Figure 3 shows that the simulated fog top tends to be larger than the observed fog top height





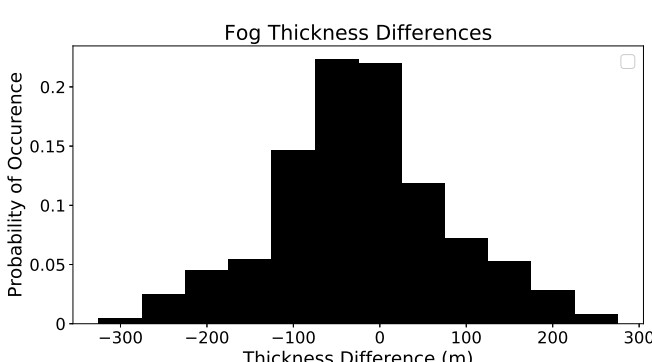

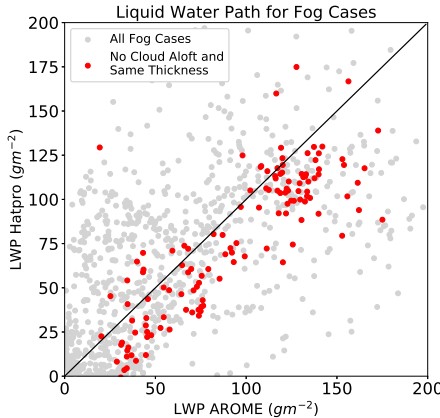

**Figure 3.** Left panel: Histogram of differences in the fog top height observed with the cloud radar and simulated by the AROME model when fog is both observed and simulated. Positive differences represent a larger observed fog top than is simulated. Right panel: Liquid water path recorded on the microwave radiometer and predicted in the AROME model for times at which both predict fog presence. All cases of fog in both model and observations without restrictions are shown in grey. Red points show LWP values where the integrated cloud thickness above the fog layer does not exceed 25 m (in either model or observation) and difference between the model and the observation fog top being less than 25 m.

with errors up to 300 m and 44 percent of differences greater than 100 m. The mean height difference is $-22.5$ m, and the standard deviation of fog top heights is 104 m.

As liquid water content is the variable responsible for causing fog, its accuracy will thus determine the quality of fog forecasts. As there are no in-situ sensors for recording the liquid water content at the observation site, the
integrated value of this, the liquid water path (LWP) from the HATPRO microwave radiometer, was used to evaluate the quality of the liquid water content forecast in the model. By comparing liquid water paths for all fog cases, we are left open to comparing not only the error in the thickness and density of the fog layer, but also of clouds aloft. Data from the radar were therefore used to select cases of fog during which the layers of cloud aloft were of less than 25 m thick. Similarly, cases where the model simulates thick clouds aloft were discarded. The liquid water path was then
compared for cases where the thickness of the fog layer predicted in the model and observed had differences of less than 25 m (figure 3). As expected, the differences in liquid water path decrease with the constraints. For cases where there is simply fog observed and simulated, the bias in LWP is $8\,\mathrm{g\,m^{-2}}$ of over prediction by the model and a standard deviation of $66\,\mathrm{g\,m^{-2}}$. For the model-observation comparisons where the fog thicknesses are the same and no cloud aloft is seen, there is a bias of $14\,\mathrm{g\,m^{-2}}$ of over prediction in the model and a standard deviation of $26.4\,\mathrm{g\,m^{-2}}$. As is
also shown in figure 3, the model more frequently over-predicts the fog thickness than under-predicts it, accounting for the positive LWP bias. Given the accuracy of the liquid water path retrieved from the microwave radiometer,





as outlined in section 2.4, of approximately $20\,\mathrm{g\,m^{-2}}$, it can be concluded that when the fog layer thickness is well predicted by the AROME model, the liquid water content inside the fog layer is also well predicted.

From the analysis presented in this section, it may be concluded that significant variations both temporally and
spatially could provide scope for the selection of a background profile which does not correspond directly to the location and time the observation was made at. The analysis of the liquid water content prediction of the model, however, shows that the model can be reliable providing that fog is forecast with a similar thickness to that observed. In the next section, the forward operator is evaluated for sources of error, and then comparisons are made between observed cloud radar profiles and profiles simulated from the AROME model. A methodology is also proposed for
selecting a background profile which better corresponds to the observed profile.

## 4   Evaluation of Observation Operator

### 4.1   Forward Operator Sensitivity Study

The radar simulator was based on radar equations which link the hydrometeor contents contained within a parcel of air to the recorded reflectivity. The attenuation and the reflectivity values both depend on the size and number of
droplets. As there are a very large number of ways a mass of water could theoretically be divided among droplets, a size distribution needs to be assumed, based on observed droplet size distributions. The droplet size distribution used in this work is consistent with the one used in the AROME model, the one-moment microphysical scheme ICE-3. This uses a modified gamma distribution, as specified in equations 5, 6 and 7. In this set of equations, N(d) is the droplet concentration. Coefficients a and b determine the mass-diameter relationship of the droplets, which
when applied to cloud droplets is well known due to their spherical nature, and are set at 524 and 3 respectively. $\alpha$, $\nu$ and $X$ are fixed coefficients of the size distribution and are set to 1, 3 and 0 respectively in ICE-3 for cloud liquid droplet. M is the liquid water content of the cell in $\mathrm{kg\cdot m^{-3}}$. As $X$ is set to 0 in ICE-3 for cloud liquid water over land, coefficient $C$ is equal to $N_0$ (the total droplet concentration), and is set to $3\cdot10^8\ \mathrm{m^{-3}}$.

$$N(D) = N_0 \frac{\alpha}{\Gamma(\nu)} \Lambda^{\alpha\nu} D^{\alpha\nu-1} e^{(-(\Lambda D)^\alpha)}, \tag{5}$$

$$\Lambda = \left(\frac{M\Gamma(\nu)}{aC\Gamma(\nu+\frac{b}{\alpha})}\right)^{(\frac{1}{X-b})}, \tag{6}$$

$$N_0 = C\Lambda^X \tag{7}$$

Microphysical observations have been investigated on fog events in previous works (Mazoyer et al., 2019; Podzimek, 1997) which tend to show lower droplet concentrations than is prescribed for continental clouds in the ICE3





microphysical scheme (of $300\,\mathrm{cm}^{-3}$ ). In the work of Mazoyer (2016), median droplet concentrations for continental

fog events ranged from $6.3\,\mathrm{cm}^{-3}$ to $147\,\mathrm{cm}^{-3}$. For continental boundary layer cloud, a study from Zhao et al. (2019) found average droplet concentrations of $320\,\mathrm{cm}^{-3}$, although this study was made in eastern China so the high droplet concentration may in part be due to elevated levels of pollution acting as condensation nuclei. Figure 4 shows the difference in distribution shapes for the modified gamma distribution when the C parameter was replaced with what was considered the reasonable lower and upper bounds of 30 and $300\,\mathrm{cm}^{-3}$.

In a review of numerous studies on low level clouds, Miles et al. (2000) attempted to find the best values for the parameters of a modified gamma distribution in order to fit the observed droplet size distributions for each of around 100 events. In this, it was found that the optimal fit warranted a significant deviation in the $\nu$ parameter attributed to each event. In this study, mean value of 8.7 was found for the $\nu$ parameter, with a standard deviation of 6.3. The reflectivity error resulting from the uncertainty of this parameter was therefore calculated with values

one standard deviation above and below the mean values. The modified distribution with these values is shown in figure 4. Uncertainties for the $\alpha$ parameter were assessed in a similar manner, for which low and high values for this parameter were taken from Thies et al. (2017) and ranged from 1 to 5. The resulting distributions are also shown in figure 4.

It can be seen from figure 4 that the effect of increasing the $\alpha$ and $\nu$ parameters was a narrowing of the distribution,

meaning fewer droplets at the smaller and larger end of the spectrum. The concentration of the largest droplet sizes (above $35\,\mu\mathrm{m}$) is therefore reduced through these changes. As the radar reflectivity is proportional to the sixth moment of the droplet size where the Rayleigh approximation is valid, this causes smaller values of reflectivity to be simulated. The perturbations in number concentration, meanwhile, were almost entirely below the value in ICE-3, with a range of $30\,\mathrm{cm}^{-3}$ to $300\,\mathrm{cm}^{-3}$ compared to a value of $300\,\mathrm{cm}^{-3}$ in ICE-3. As may be seen in figure 4, this

caused an increase in the number of large droplets (over $50\,\mu\mathrm{m}$ and thus an increase in the simulated reflectivity).

In order to assess the uncertainty in the simulations resulting from the uncertainty in the size distribution parameters $\alpha$, $\nu$ and $C$, simulations were made by perturbing these parameters according to the typical uncertainties from the literature previously discussed. An atmospheric profile under fog conditions was selected from the AROME model with a maximum LWC of $0.12\,\mathrm{g m}^{-3}$ at 71m agl. Reflectivity was then simulated with changes to the de-

fault parameters of the modified gamma distribution. Firstly, the number concentration was held constant, whilst perturbations were made to the $\alpha$ and $\nu$ parameters. The same process was repeated, keeping values of $\alpha$ and $\nu$ constant and simulating the reflectivity with perturbations in the number concentration. The obtained distribution of reflectivity values is shown in figure 5. It can be seen that the uncertainty in the number concentration contributes the largest to the uncertainty in the simulated reflectivity. For the altitude at which the liquid water content is the

largest, at $0.12\,\mathrm{g m}^{-3}$ the reflectivity difference reaches $10\,\mathrm{dB}$ between the highest and lowest readings, and $4\,\mathrm{dB}$ between the 25[th] and 75[th] percentile. For the changes in the $\alpha$ and $\nu$ parameters, the difference between the highest and lowest reading is $8\,\mathrm{dB}$, with a difference of only $1\,\mathrm{dB}$ between the 25[th] and 75[th] percentiles. It can be noted that

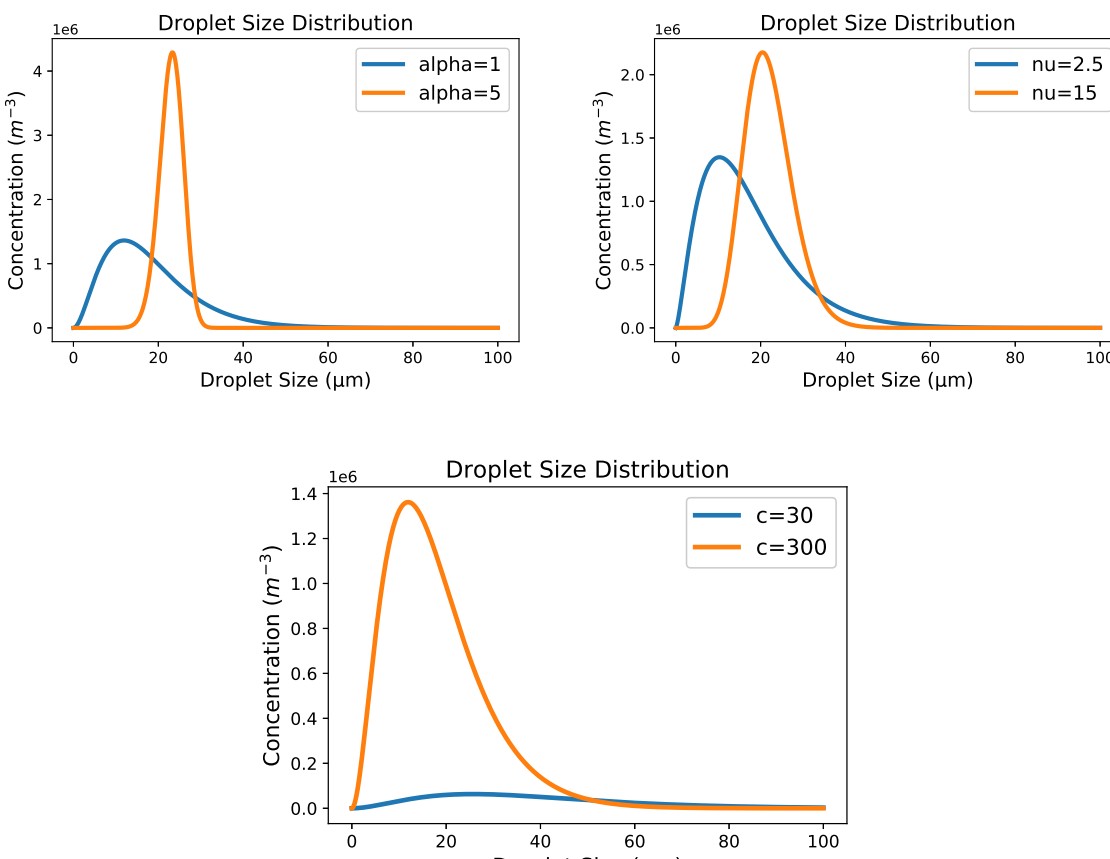

**Figure 4.** Modified gamma distributions for a liquid water content of $0.12\,\mathrm{g\,m^{-3}}$ prescribed by the ICE-3 scheme. Top left to bottom right: with $\alpha = 1$ and $\alpha = 5$ (default $= 1$); with $\nu = 2.5$ and $\nu = 15$ (default $= 3$); with $C = 30\,\mathrm{cm^{-3}}$ and $C = 300\,\mathrm{cm^{-3}}$ (default $= 300\,\mathrm{cm^{-3}}$).

the perturbations in the $\alpha$ and $\nu$ constants were mainly above the values used in the ICE-3 microphysical scheme, with ranges of 1 to 5 and 2.5 to 15 respectively, compared to their values in ICE-3 of $\alpha = 1$ and $\nu = 3$.

The reflectivity simulated from the default parameters in ICE3 can be seen from the plots as the minimum reflectivity simulated in where changes are made to number concentration, and the maximum in the distribution with changes to $\alpha$ and $\nu$. When the $25^{\text{th}}$ to $75^{\text{th}}$ percentiles are considered, the total uncertainty in the simulated reflectivity caused by the uncertainty of the three parameters may be evaluated to be $6\,\mathrm{dB}$.

The results of the microphysics study highlights that non-negligible errors on the simulated radar reflectivity can
be attributed to errors in the fixed parameters of the droplet size distribution. The $\alpha$ and $\nu$ parameters were found to contribute to the errors to a lesser extent than the droplet concentration number.





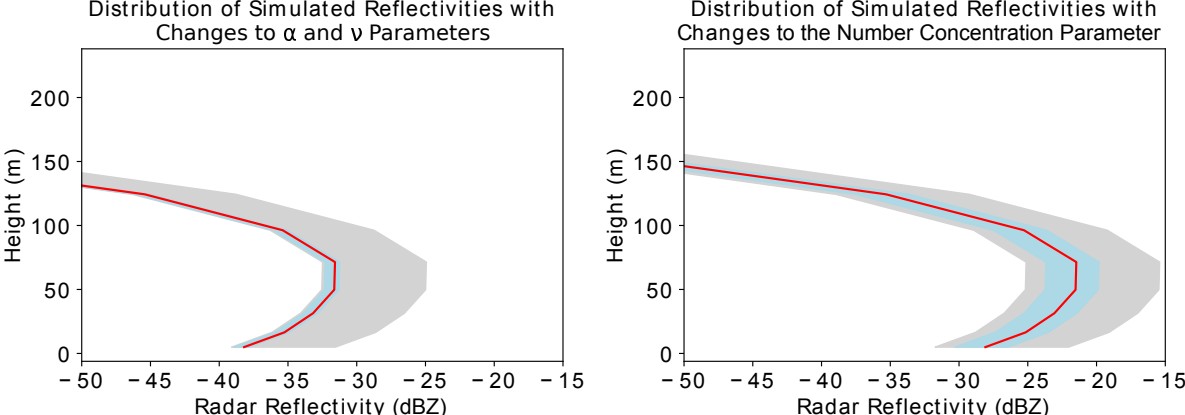

**Figure 5.** Spread of simulated values of radar reflectivity with height for a change in the microphysical parameters of a modified gamma distribution: $\nu$ and $\alpha$ (left panel) and number concentration (right panel) for a fog profile. The 25[th] and 75[th] percentiles are shown in blue, and the median reflectivity shown by the red line.

### 4.2 Most Resembling Profile (MRP) Selection Method

Section 3.3 has demonstrated that significant errors are seen both spatially and temporally in the AROME model when corresponding exactly to the time and location of the observation. In order to improve the accuracy of the
background profile, a method was thus devised to select the model profile which best corresponds to the measured atmospheric profile. For this, reflectivity for all profiles throughout the domain was simulated for a time window of 6 hours ($\pm 3$ hours). Reflectivity differences were then found between the observed profile and each of the simulated profiles. The weighted RMSE was then found from equations 9 and 8. The profile with the smallest weighted RMSE was selected as the 'most resembling profile'. This method is similar to the most resembling column (MRC) method
used by Borderies et al. (2018) to calibrate and validate the RASTA cloud radar observation operator. It also includes an altitude-dependent weighting function (equation (8)) as was used in Le Bastard et al. (2019), which puts a larger weight on the bins at a lower height. In this equation, Height is the height of the reflectivity bin and Altmax is the maximum altitude considered which for this study was set to $5000\,\mathrm{m}$.

$$W_i = \frac{2}{\frac{Height_i}{\mathrm{Altmax}} + 1} - 1 \tag{8}$$

Weighted RMSE $= \sqrt{\dfrac{\sum_{i=0}^{i=Maxlev} W_i (Z_{Observation} - Z_{Simulation})^2}{n}}$     (9)

    Using the MRP selection, the simulated reflectivity can be improved with the choice of a more appropriate background profile. This is often the case when fog is predicted by the model, but none is seen, in which case it





is generally possible to select a clear-sky profile. The method is also able to deal with temporal shifts in the fog event between the model and observations, as well as differences in the vertical structure. Figure 6 illustrates the

MRP selection during a fog event observed at SIRTA on the 22 November 2018. It demonstrates well how much benefit is brought by the selection method with fog structures closer to the observation. In both the observation and simulation, stratus lowering events were seen, however, the model predicted the event to occur around 90 min before it was observed, and the fog top height to wrongly increase from 200 m to 400 m between 10:00 and 11:00 UTC. This is also shown in 7, for which the correction in fog top height and values of simulated reflectivity is clearly

illustrated on a specific vertical profile selected during the fog mature phase. The stratus was also predicted to lower from 100 m over one hour in the model, which was corrected to lower from 250 m over two hours with the MRP selection method. The MRP selection method was able to select background profiles to rectify temporal errors at the fog formation but also the fog vertical structure.

### 4.3  Contoured Frequency by Altitude Diagrams

In order to investigate the capability of the forward model to reproduce the overall structure of observed reflectivity, Contoured Frequency by Altitude Diagrams (CFADs, Yuter and Houze, 1995) calculated both from the observations and the simulations were compared in figure 8. In these figures, the number of cases in each radar reflectivity bin and each altitude level are shown between 50 m to 1000 m with a bin width of 1 dB. The distributions at each height level were then normalised, and the relative frequency of each bin is shown on the plots. The CFADs were plotted

using data for which reflectivity at each range gate was obtained in the observation, nearest corresponding profile and the MRP.

In the observations, the reflectivity in the lower 300 m is most concentrated between −30 dBZ to −20 dBZ and becomes gradually less concentrated at lower reflectivities. This contrasts the nearest corresponding profile simulations where there are significantly fewer radar reflectivities below −30 dBZ, and a concentration of higher values around

−25 dBZ. This distribution is improved by the implementation of the MRP method, where a more even distribution of reflectivities may be seen in the bottom 200 m. Though the distribution of simulated reflectivity generally improves by using the MRP method, a large concentration of values between −23 dBZ and −20 dBZ persists which is not seen in the observation CFAD.

### 4.4  Statistics on Reflectivity Innovations

For the period for which the fog classification was previously applied to, between November 2018 and February 2019, radar reflectivity was simulated for the 28 km by 28 km domain for the entire period, after which the MRP method was applied. The observations were downscaled to the resolution of the simulations by using the observation which corresponded most closely to the time of the simulation, and by using the bin corresponding most closely to the level heights of the model.

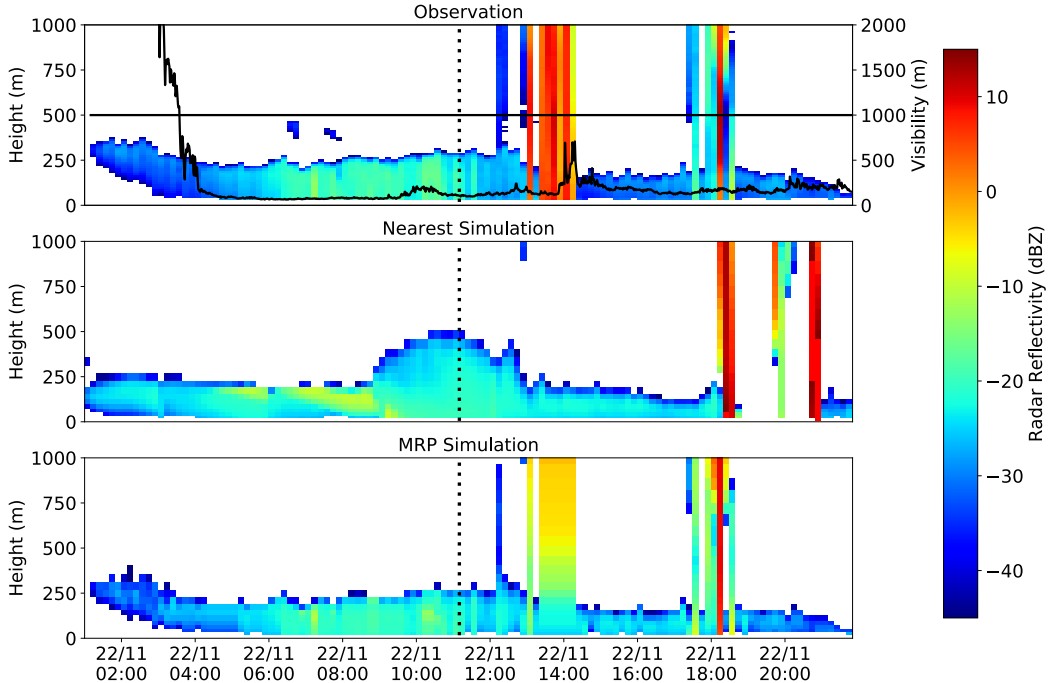

**Figure 6.** Radar reflectivity and surface level visibility from a fog event at SIRTA observed on 22/11/2018 (top), simulated from the nearest gridpoint (middle) and with the MRP selection method (bottom). The dashed black line indicates the time at which the following plot of reflectivity profiles is taken. These plots show the reflectivity for all profiles, some of which were later removed due to significant rain content. Time is in UTC.

The radar simulator relies on the Mie approximation to derive the radar reflectivity. This approximation is valid for uniform isotropic particles, which may be assumed for liquid cloud droplets. However, for snow, graupel, ice and rain, whose shape can be significantly more complex, this approximation can no longer be assumed to be valid, and larger errors of simulated reflectivity are likely to be caused by this. It was therefore decided to limit this study to reflectivity differences only due to the hydrometeors which are mainly responsible for fog in the mid-latitudes

in winter: liquid water droplets. For the observation, a mask proxy was provided by the developers of the BASTA instrument to classify the hydrometeor type. The mask was used to reject from the statistical analysis cloud radar observations containing rain, drizzle and ice below 200 m in the observations.

In the model space a mask based on simulated reflectivity was used to discern whether rain, ice, snow or graupel significantly contributed to the simulated reflectivity. This was made by finding reflectivity differences between



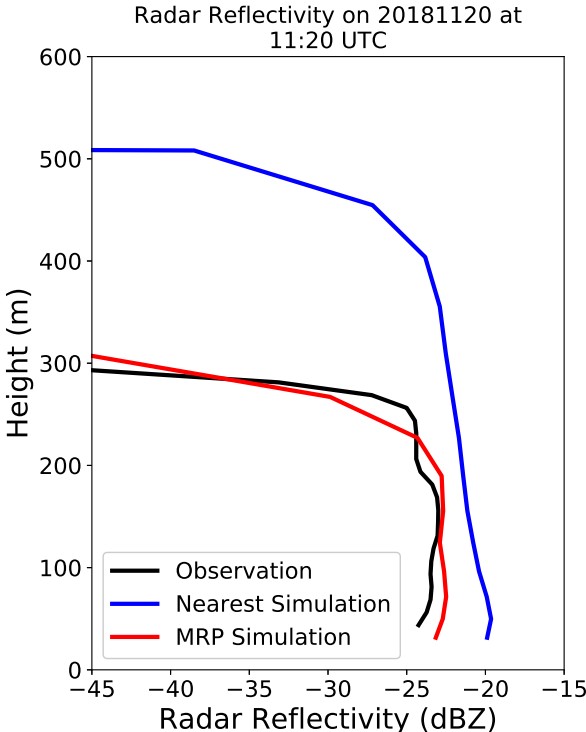

**Figure 7.** Radar reflectivity profiles of the observation, simulation from the nearest gridpoint, and MRP simulation, from the mature phase of the fog event at SIRTA on 22/11/2018. At this point in the fog event, the model overestimated the thickness of the fog layer by around 33 %

the simulations containing all hydrometeors and the simulations for only cloud liquid water. Profiles containing significant reflectivity differences (of greater than 3 dB) were masked. This value was chosen as a 3 dB increase in radar reflectivity corresponds to a doubling of the received power. This effectively means that where differences between radar reflectivity simulated with only liquid water and radar reflectivity simulated with all hydrometeors exceeds 3 dB, the other hydrometeors contribute more to the radar reflectivity than liquid water content. Due to the

effect of attenuated signal which occurs when the radar signal passes through a rain event but impacts the readings above as well as inside the rainy atmosphere, where rain was found below 200m, the entire profiles were also removed from the statistical calculations.

Innovations (the difference between observed values and simulated values) were then calculated with the simulations for the nearest corresponding gridpoint and the MRP selection method. For these calculations, data was only

used for which the range gate in both the simulation and observation had reflectivity signal above the sensitivity of the instrument. Figure 9 shows the standard deviation and bias at each height level. Statistics are shown up to 1200 m altitude as above this height, not enough cases without significant impact from ice can be selected. It may be seen





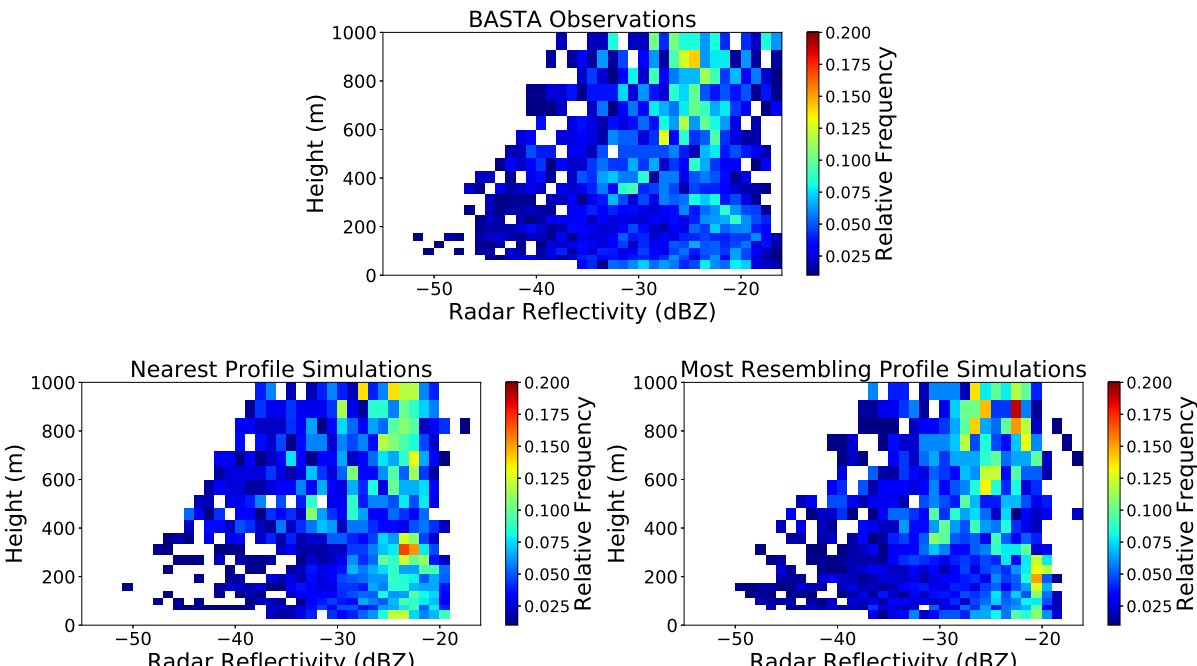

**Figure 8.** CFADs of reflectivity observed and simulated from the nearest corresponding and most resembling profiles for the period 01/11/2018 to 19/02/2019 where cloud is seen in all three frames.

from the plots that both the bias and standard deviation are reduced at almost all heights with the implementation of the MRP method. The standard deviation was highest for the nearest profile at a height of 80 m agl, for which

the standard deviation was 12.6 dB. The MRP selection method was able to reduce this value to 4.7 dB, showing an improvement of 7.9 dB. Between 400 m and 1000 m, the bias for the nearest profile was between 4.7 dB and 6.2 dB. For the MRP, it remained below 1.5 dB for the same height range. It may also be seen that using the MRP method increases the count and hence more retrievals may be made with this method compared to the nearest gridpoint method. This study shows that, after removal of the largest background errors, the forward operator used in this

study is able to correctly simulate the radar reflectivity observed during fog conditions. For the application of future 1D-Var retrievals and data assimilation, this brings the benefit of the simulations not needing to be bias-corrected. The reduction in the standard deviation may also improve the accuracy of the retrieved profiles.

     Additionally, data assimilation relies on the assumption that the distribution of background and observation errors are Gaussian. Though in real-world scenarios, a perfectly Gaussian distribution is rarely observed, certain manual

and statistical checks may be made to ensure that a distribution is approximately Gaussian. According to Bulmer (1979), one of these checks is for the skewness and excess kurtosis of a distribution to be between −1 and 1. Figure 10 shows the distribution of innovations both for the co-located profile and the MRP profile at 80 m altitude. For the nearest profile, the Gaussianity is not satisfied, with values of skewness and excess kurtosis of 0.53 and 1.196





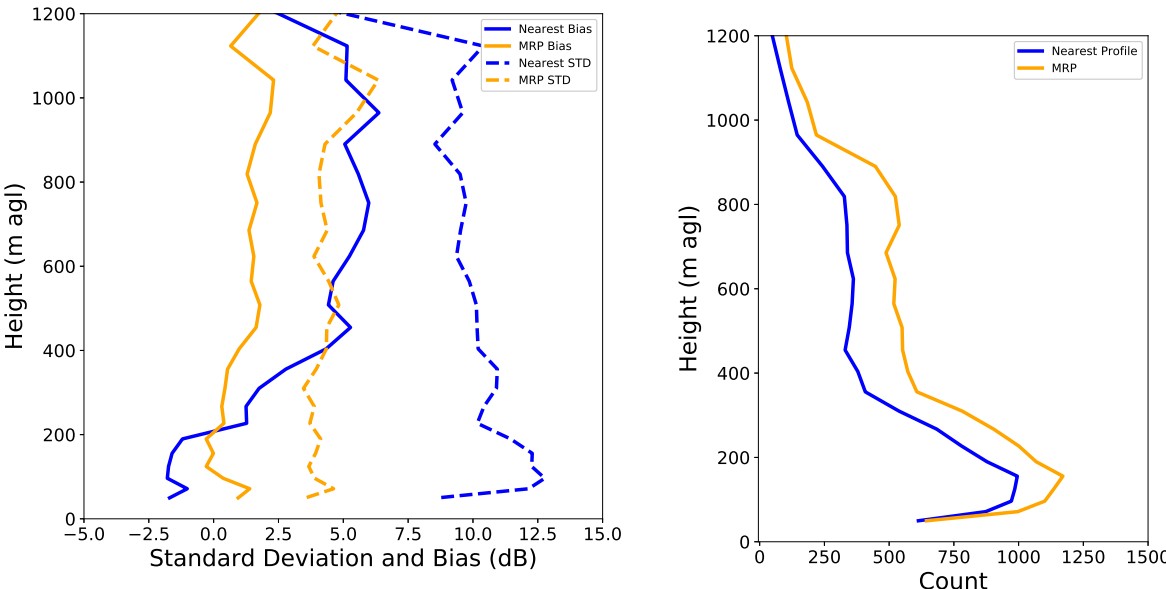

**Figure 9.** Left panel: The bias and standard deviation of Observation - Simulated radar reflectivity at SIRTA for the period 01/11/2018 - 19/02/2019. The statistics were calculated for instances when reflectivity was both observed and simulated at a given range gate at a given time. Right panel: the count of cells for which reflectivity was observed and simulated.

respectively. The MRP method did not satisfy this criteria either, with values of 0.68 and 2.68 respectively. This problem was due to the fact that more data was seen in the extremes of the distribution, with a reduction in the reflectivity differences for many cases but some cases not being improved, for example when fog was not forecast at all throughout the domain. In order to rectify this, the most extreme 10 % of data points corresponding to the simulated errors above 16 dB for the nearest profile selection and 6.5 dB for the MRP were removed. After this data screening, the excess kurtosis for the nearest profile and MRP were reduced to 0.68 and 0.64 respectively demonstrating that distributions of innovations can be safely considered as Gaussian for future data assimilation steps. These conditions were also met for the distributions at higher levels (not shown).

## 5 Conclusion and Discussion

In preparation of future data assimilation of newly developed 95 GHz cloud radar observations, this work aimed at better understanding uncertainties associated with background, observations and forward operator errors during fog events. An overview of fog forecast errors was firstly made using an instrumental dataset from SIRTA, Paris during winter 2018-2019. It was concluded that the AROME model tends to over-forecast fog, with 1.6 times the amount



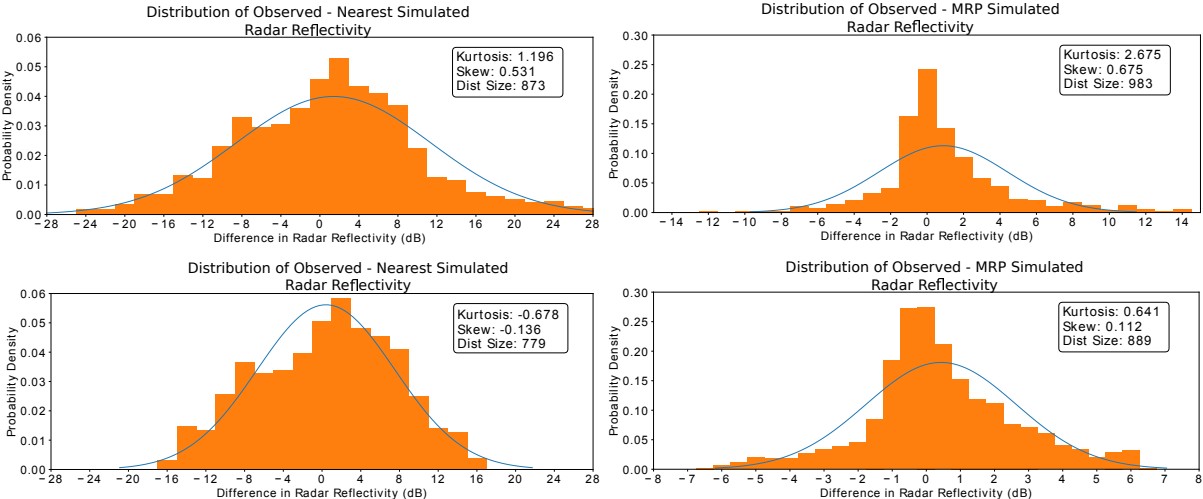

**Figure 10.** From top left to bottom right: distribution of observed minus simulated reflectivity errors for the nearest corresponding profile when no data has been excluded, the MRP when no data has been excluded, the nearest corresponding profile when 10% of data has been excluded and the MRP when 10% of data has been excluded. All distributions are shown at 80 m agl, for reflectivity innovations when there is signal both in the simulation and observation above the sensitivity threshold. The blue line shows the Gaussian distribution with the same mean and standard deviation.

of fog profiles being forecast compared to those observed over the investigation period. It was also shown that the model tends to over-forecast the fog top height, and that fog forecasts are prone to temporal errors up to three hours. Fog presence was also shown to display significant spatial variation. However, for times in which the fog top height was well predicted by the model, the liquid water path was also well predicted, with a standard deviation in LWP difference of $26.4 \, \mathrm{g \, m^{-2}}$ when the fog top height had a difference of less than $25 \, \mathrm{m}$.

In order to correct for modelling errors, a method for selecting the model profile which best resembles the observed profile was proposed. This contained a weighting function to ensure that the selected profile is optimised for fog, in case there were also clouds aloft in the observed profile.

As previously discussed, variational retrieval methods assume un-biased and normally distributed background and observation errors. In order to assess whether these conditions were met, statistics of the differences between observations and simulated reflectivity were calculated for both the nearest corresponding profile and the MRP. It was found that whilst there was a significant bias for the nearest corresponding profile ($-2 \, \mathrm{dB}$ to $5 \, \mathrm{dB}$ below $1000 \, \mathrm{m}$) this was greatly reduced for the MRP ($0 \, \mathrm{dB}$ to $1.5 \, \mathrm{dB}$ below $1000 \, \mathrm{m}$). The standard deviation was also reduced from $10.1 \, \mathrm{dB}$ to $4.7 \, \mathrm{dB}$ at $200 \, \mathrm{m}$ through the implementation of the MRP method. When testing the distributions for normality, it was necessary to exclude $10 \, \%$ of the data (limiting the innovations to $-17 \, \mathrm{dB}$ to $17 \, \mathrm{dB}$ for the nearest profile selection method and $-6.5 \, \mathrm{dB}$ to $6.5 \, \mathrm{dB}$ from the MRP method) in order for the excess kurtosis requirements to be met.



The contribution of uncertainties in the radar simulator due to assumptions on the droplet size distribution was

also analysed. The uncertainty due to shape parameters of the cloud droplet size distribution was assessed to be 6 dB. Although this value seems large considering that the standard deviation of innovation errors was reduced to less than 5dB with the MRP method, the use of a 2-moment microphysical scheme, such as LIMA (Vié et al., 2015), which is currently being tested for operational use, promises to reduce this error by a prognostic evolution of the droplet number concentration.

The results shown here indicate the suitability of the method for future 1D-Var retrievals of liquid water content profiles from the BASTA cloud radar by using an appropriate background profile from the AROME model and a consistent radar simulator. The benefits of this could be seen through the assimilation of the retrieved profiles into a high resolution model as well as deriving continuous measurements of the liquid water content profile throughout the boundary layer, which would be of particular use to fog process studies. When background errors are reduced, the

radar simulator was also found to be suitable to simulate the BASTA cloud radar reflectivity during fog conditions paving the way for larger model evaluations during fog events.

*Data availability.* The AROME forecasts are available on request on the website https://donneespubliques.meteofrance.fr/. Data used from SIRTA is publicly available from http://sirta.ipsl.fr/. The cloud radar data was an updated version of that publicly from the SIRTA website, and requests for this can be made to julien.delanoe@latmos.ipsl.fr

*Author contributions.*

AB performed the analysis documented in the paper. PM, OC and BV supervised this analysis. JD provided the cloud radar data and relevant assistance. JCD provided the data from SIRTA. MB provided the radar simulator.

*Competing interests.* The authors declare that they have no conflict of interest.

*Acknowledgements.* This work has been funded by the French ANR SOFOG3D (SOuth west FOGs 3D experiment for processes

study, ANR-18-CE01-0004). We extend our acknowledgments to the technical and computer staff of SIRTA Observatory for taking the observations and making the data set easily accessible.





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
