# Peer review of "W-band Radar Observations for Fog Forecast Improvement: an Analysis of Model and Forward Operator Errors"

_Atmospheric Measurement Techniques, 2020_

## Referee Comment (RC1)

Review of Bell et al.: "W-band Radar Observations for Fog Forecast Improvement: an Analysis of Model and Forward Operator Errors"

This paper presents an analysis of model and forward operator errors when assimilating 95 GHz reflectivity profiles to improve fog forecasts. This is a very interesting approach, which follows from earlier work from Borderies et al. work. The paper is very well written and structured. I do have two main comments, which I believe need to be addressed before the paper can be accepted, and they both relate to the same general issue. Then I only have a list of relatively minor comments.

Major comment:
My only major issue with the paper is related to how the assimilation of W-band reflectivities will actually look like in practice. The current version of the paper confused me in that respect, mostly sections 4.1 and 4.2. I was a bit confused first in section 4.1. What is the purpose of this exercise in the authors' mind ? I though initially that you wanted to demonstrate that the information content of reflectivity profiles to constrain assumptions in the three drop size distribution parameters was high, and you do show that the simulated reflectivity profiles do change substantially with the droplet concentration. In the retrieval world, such large change in the reflectivity profile as a function of a free parameter is exactly what you hope for. But you don't present things like that, you talk about this result as an "uncertainty". So here's my problem, which goes back to the main objective of this study, which is to demonstrate the potential impact of assimilating reflectivity. How is that going to work? What are going to be the increments produced by the reflectivity constraint? I thought you would free up some of the DSD parameters to adjust them to the observed reflectivity profile. But you don't seem to plan to do that, otherwise you would not treat this variability as an error of your forward model.
What I think this means is that the paper should include some description of how the assimilation is going to work.
I reached the same conclusion when reading section 4.2, explaining the MRP process. I am a bit worried by this approach, but maybe I shouldn't, as it depends how you are going to use that in practice. I thought initially that this was a process to shift the whole time series of model profiles by a given amount, so that you can take into account a possible mismatch in time with the initial development of the fog layer. But here you take any profile within a 6h window that matches the observations, which means that you completely lose the spatial continuity of the simulated fog layer in time, it can be any point of the 28*28km domain and it will change from one time step to another. How is that going to help in real situations when you assimilate Z profiles?
Again, in order to put your very interesting results in perspective of how you are going to use them when assimilating the reflectivity profiles for real, you need a section or a paragraph explaining how it's going to work.

Minor comments:
1. Lines 29-30: you say that this has been increasingly discussed "in recent years" but you cite a paper from 11 years ago, so it reads a bit funny. Any more recent discussions to justify your claim?
2. Line 41 : what is "advance time" ? I suspect you talk about "lead time" here ?
3. Line 42: I suggest "developed" instead of "seen"
4. Line 55-56 : what lead time of simulation are you using for this ?
5. Lines 60-61: why is the forward approach better posed than the Z - LWC conversion ("backward approach"_ ? The problem with Z to LWC is when there are few drizzle-sized drops in the volume, but I assume that in fog layers this is not so much of an issue, so the

Z to LWC approach trained with observations may be as accurate as the other way around. Please clarify why you think that's the case.

6. Title of section 2.2 : should BASTA be in capital letters ?
7. Line 102 : Fourth (not forth)
8. Lines 108-109: it's a bit inaccurate, I would rather say : "the fact that BASTA has separate transmitting and receiving antennas ..."
9. Paragraph starting line 200: I think it would be useful to remind the reader what is expected to be the perfect score for each index used and where you consider that skills are not satisfying.
10. I have a suggestion about this analysis of scores. Showing just scores seems a bit incomplete and missed opportunity to dig more into the model performance. For instance, I would have liked to see PDFs of visibility from the model when fog is observed. It could be that the model is very slightly underestimating or overestimating visibility, which could change the scores substantially when using a hard limit of max visibility of 1km. We need to know how well the model predicts visibility overall in observed fog conditions. That would also provide quantitative information to the modellers to improve parameterizations. Also, because you have different kinds of fog, would you have enough cases to provide those scores and visibility PDFs for different types of fog you introduced earlier?
11. Line 211-212: I don't believe that's what a CSI of 0.32 means. This is how to interpret POD, not CSI. This needs to be addressed.
12. Figure 1 and associated text: While showing an example (good idea), it would be interesting to add two panels with vertical cross-sections of observed and modelled fog at the SIRTA site as well. Later, once you have developed your matching technique you could show the result on the same vertical cross-section to demonstrate the improvement visually as well.
13. Line 237: " 12-hour window". Isn't that too big a window ? Atmospheric conditions can change substantially, as well as radiative forcing over such a large window. As a result, you could get fog in the model developing from very different processes.
14. Lines 247-248: it is very interesting to learn that the model tends to produce shorter fog episodes. Is that only when fog episodes are both observed and simulated or separately for all observed and all simulated ? The reason why I ask is because you say that the model tends to produce a lot of cases that are not observed. What would that number be for the simulated when observed and simulated when not observed, any difference ?
15. Figure 3 and its interpretation. I think your statement that simulated fog top tends to be larger (and you should say higher not larger here) is not obvious from the figure. I'd rather stick to the bias value, maybe say that distribution of differences (is it model – obs by the way ?) is centred on zero with the std of 104m.
16. Paragraph starting line 336. Those are very interesting results, as is the impact on reflectivity from Fig.5. However, maybe some combination of the possible values of these three parameters is unrealistic. Does the literature say anything about the co-variability of these parameters? For instance are high concentrations always associated with a narrower range of nu or the other parameter ? That would help build a more realistic picture of the true variability of the reflectivity simulations if you can use such knowledge in your sensitivity analysis.
17. Line 377: " occur around 90 min". Is that your visual inspection estimate or is it the value that your MRP process found ?
18. Line 379: "Figure 7"

19. Figure 6: I think you should mask out the profiles with precipitation in both observations and model (using grey squares or something), as it is distracting from the main message of the figure.
20. Line 444 and Figure 10: One thing you should mention from this Figure, which is very positive, is that the distribution of errors is a lot narrower and around zero when MRP is used.

Good luck with the review,
Alain Protat
Melbourne, 29/01/2021

---

## Author Comment (AC1)

**Manuscript Review: W-band Radar Observations for Fog Forecast Improvement: an Analysis of Model and Forward Operator Errors**

Thanks to all of the reviewers who kindly gave their time to analyse the article and returned relevant comments to help clarify the research for future readers. We hope to respond to your queries below.

Reviewer 1

**Comment:** My only major issue with the paper is related to how the assimilation of W-band reflectivities will actually look like in practice. The current version of the paper confused me in that respect, mostly sections 4.1 and 4.2. I was a bit confused first in section 4.1 . What is the purpose of this exercise in the authors' mind ? I though initially that you wanted to demonstrate that the information content of reflectivity profiles to constrain assumptions in the three drop size distribution parameters was high, and you do show that the simulated reflectivity profiles do change substantially with the droplet concentration. In the retrieval world, such large change in the reflectivity profile as a function of a free parameter is exactly what you hope for. But you don't present things like that, you talk about this result as an "uncertainty". So here's my problem, which goes back to the main objective of this study, which is to demonstrate the potential impact of assimilating reflectivity. How is that going to work? What are going to be the increments produced by the reflectivity constraint? I thought you would free up some of the DSD parameters to adjust them to the observed reflectivity profile. But you don't seem to plan to do that, otherwise you would not treat this variability as an error of your forward model.

**Response:** Firstly I would like to clarify that although developments are being made in the AROME model to have a two moment microphysical scheme (which would allow prognosed number concentration number to be specified in the background profile) currently it is being run operationally with a one moment scheme. For a simulation of radar reflectivity from model output, all initial simulations will thus contain uncertainty due to the uncertainty in the cloud microphysics which cannot be prescribed.

You indeed raise an important point which was not emphasised enough in the paper – that the sensitivity of simulated radar reflectivity to the microphysics means that it is possible (at least in theory) to perform retrievals on the number concentrations (and potentially other microphysical parameters) as well as the liquid water content. For 1D-Var future retrievals it will be an aim to make retrievals on the droplet number concentration.

As the approach that we are looking towards is a 1D+3D-Var method, this would allow making retrievals for number concentration in the initial stage, but not using this information for the 3D-Var data assimilation (as only a 1-moment microphysical scheme is used in the model). In order to do this, the background profile would also have to include the droplet number concentration, and a background error covariance matrix would have to be computed to include this variable. It is likely that by including the number concentration in the retrieved variables in the 1D-Var, that the accuracy of LWC retrievals will be improved, and so this could be a benefit for assimilating with the indirect approach. However, by introducing the droplet number concentration, an additional degree of freedom is also introduced, with no new information, which could also degrade retrievals of LWC. Also, if a direct 3D-Var approach was used, it would not be possible to use this information as microphysical variables cannot be part of the control variables. It is thus important for us to start by validating the approach that could be implemented operationally in AROME and potentially in a second stage evaluate more advanced approaches that could be also used with 2-moment microphysical scheme.

**Change:** Inserted paragraph in introduction: 'The main goal of this work with respect to future OE retrievals is to use radar reflectivity observations in combination with microwave radiometer brightness temperature observations to provide estimations of liquid water content in addition to temperature and humidity. As radar reflectivity is mainly sensitive to the total cloud droplet concentration and the size distribution of the droplets, it may also be possible to add parameters related to this to the set of retrieved variables in an OE algorithm. However, as a one moment microphysical scheme is currently used in the operational AROME model, and due to the added complexity of adding the droplet concentration number, first 1D data assimilation experiments will focus only on the liquid water content retrieval.

Added line 505 in conclusions: 'Future methods of OE retrieval with cloud radar could also include the droplet concentration and size distribution parameters in the set of variables to be retrieved. In this case, uncertainties from microphysical assumptions could be greatly reduced. Indeed, the significant sensitivity of the radar simulator towards droplet size distribution properties, as shown in this study, could prove to be advantageous for retrievals of these properties. The need for a background covariance matrix to include the additional variables, as well as a lack of additional observations which could constrain the retrieval, means that this would, however, add additional complexity.

**Comment:** I reached the same conclusion when reading section 4.2, explaining the MRP process. I am a bit worried by this approach, but maybe I shouldn't, as it depends how you are going to use that in practice. I thought initially that this was a process to shift the whole time series of model profiles by a given amount, so that you can take into account a possible mismatch in time with the initial development of the fog layer. But here you take any profile within a 6h window that matches the observations, which means that you completely lose the spatial continuity of the simulated fog layer in time, it can be any point of the 28*28km domain and it will change from one time step to another. How is that going to help in real situations when you assimilate Z profiles? Again, in order to put your very interesting results in perspective of how you are going to use them when assimilating the reflectivity profiles for real, you need a section or a paragraph explaining how it's going to work.

**Response:** The approach we aim to take to begin with is a 1D-Var + 3D-Var approach. Retrievals will be made for profiles of LWC, which can then be assimilated into the model. With this approach, the selection of background profiles in a non-chronological order should not be a problem as each profile will be modified according to the observation. We expect larger problems in the 1D-Var convergence and accuracy if the background LWC profile is significantly different to the observation, which happens fairly regularly if the nearest model profile is taken. For a direct 3D/4D-Var assimilation of radar reflectivity, a more careful method of selecting background profiles may be needed. One way to resolve this may be by putting constraints on the temporal and spatial separation between background profiles used with temporally adjacent observations (i.e. within a maximum distance and time range).

**Comment:** you say that this has been increasingly discussed "in recent years" but you cite a paper from 11 years ago, so it reads a bit funny. Any more recent discussions to justify your claim?

**Change:** We agree, two other references were added from last six years (Hu et al., 2019; Wilkzac et al, 2015)

**Comment:** Line 42: I suggest "developed" instead of "seen"

**Change:** As suggested

**Comment:** what is "advance time"? I suspect you talk about "lead time" here ?

**Change:** Thanks for pointing out the inaccuracy. This has been changed to 'forecast term', which refers specifically to the time difference between the analysis and the time of a forecast phenomena, whereas 'lead time' refers to length of time between the issuance of a forecast and the occurrence of the phenomena that were predicted.

**Comment:** Why is the forward approach better posed than the Z - LWC conversion "backward approach"? The problem with Z to LWC is when there are few drizzle-sized drops in the volume, but I assume that in fog layers this is not so much of an issue, so the Z to LWC approach trained with observations may be as accurate as the other way around. Please clarify why you think that's the case.

**Response:** Compared to a simple power law Z-LWC relation, the forward model approach has the benefit of adapting many microphysical parameters. It was shown in the PhD thesis of Waersted (2018) by deriving coefficients for the power law from observations of fog microphysical properties, that the coefficients needed for such a law can vary significantly between different fog cases. There is a lack of in-situ measurements to be able to well define these Z-LWC relations within fog and we have to rely with empirical laws based on other cloud types.

[Figure]

**Figure 1:** (taken from thesis of Waersted, 2018) Z-LWC relations from commonly cited studies and found from in-situ cloud microphysics data as part of a study for the thesis of E.Waersted.

The forward model which was used in this study also takes attenuation (both from hydrometeors and gases) into account which is generally agreed to be more difficult to model in the backwards direction. Uncertainties are also easier to control and model in the forwards direction (Reitter et al., 2011).
I will also mention briefly, since you mention here about drizzle droplets, that surprisingly during fog events with large liquid water contents in the model space, significant amounts of rain content would be modelled in the same grid point (in the AROME model, there is no 'drizzle' species of hydrometeor, just cloud liquid water and rain). Significant here refers to the impact it has on simulated reflectivity and is discussed in section 4.4 of the article. Though we do not initially intend to make retrievals for times when drizzle is present, using a forward model being able to easily take into account mixed-phase clouds is likely to improve retrievals for these cases.

**Change:** Added**: '**The main advantage of using a forward model compared to a backward model, when only cloud droplets as hydrometeors are considered, arises from the ability to easily model attenuation from cloud droplets, water vapour and dry air in the forward direction.'

**Comment:** Title of section 2.2 : should BASTA be in capital letters ?

**Change:** As suggested

**Comment:** Line 102 : Fourth (not forth)

**Change:** As suggested

**Comment:** Lines 108-109: it's a bit inaccurate, I would rather say : "the fact that BASTA has separate transmitting and receiving antennas

**Change:** As suggested

**Comment:** Paragraph starting line 200: I think it would be useful to remind the reader what is expected to be the perfect score for each index used and where you consider that skills are not satisfying.

**Change:** Added: 'FBI scores can range from zero to infinity, where a perfect score is one, and less than one indicates an under-prediction of events and greater than one indicates an over-prediction. CSI scores can range from zero to one, with the perfect score being one.'

**Response:** Regarding 'satisfying skills'- often cost benefit analysis (how the forecast is used vs cost of improving forecast) is performed to analyse acceptable scores. Although the most benefit of the fog forecasts is likely to be to airports/airline companies, again I think to break down the costs of grounding aircraft would be beyond the scope of this article.

**Comment:** I have a suggestion about this analysis of scores. Showing just scores seems a bit incomplete and missed opportunity to dig more into the model performance. For instance, I would have liked to see PDFs of visibility from the model when fog is observed. It could be that the model is very slightly underestimating or overestimating visibility, which could change the scores substantially when using a hard limit of max visibility of 1km. We need to know how well the model predicts visibility overall in observed fog conditions. That would also provide quantitative information to the modellers to improve parameterizations. Also, because you have different kinds of fog, would you have enough cases to provide those scores and visibility PDFs for different types of fog you introduced earlier?

**Response:** It's an interesting point, and indeed there are many things that could be further developed when analysing the model's ability to forecast fog events. I've included a histogram (Figure 2) showing the distribution of visibilities from AROME and the visibility monitor. You can see that, as we know, AROME predicts more instances of visibility under 1km. However, we need to go above a visibility of 1400m to see the frequency of observations for that visibility range increase above the frequency of the model diagnostic for the same visibility range. The false alarm ratio was calculated again by defining fog with a visibility of firstly 800m and then 1200m. The change was +1% when it was reduced to 800m and -1% when it was increased to 1200m. It was thus concluded that although the visibility diagnostic may not be perfect, the statistics would not change drastically if it turns out that visibility is slightly over- or under-estimated.

[Figure]

**Figure 2:** Histogram of observed and modelled visibilities for winter 2018/2019 at SIRTA.

The visibility diagnostic is fairly new and work will be done to analyse its accuracy, but this is beyond the scope of this paper. Following the recent fog campaign (SOFOG3D), a more in depth analysis of the AROME model scores is being performed by other researchers at CNRM (Dedicated PhD thesis of Salomé Antoine).

**Comment:** Line 211-212: I don't believe that's what a CSI of 0.32 means. This is how to interpret POD, not CSI. This needs to be addressed

**Response**: In our opinion, it is consistent with the definition from Ebert (2008), who writes: 'The TS, also known as the critical success index, gives the fraction of all events forecast and/or observed that were correctly diagnosed'. The CSI can be an informative index to include in addition to the POD, as it also takes into account the over-prediction by the model.

**Change**: Added: 'As the CSI "assumes that the times when an event was netiher expected nor observed are of no consequence" (Schaefer, 1990) this can be a useful metric to consider.'

**Comment:** Figure 1 and associated text: While showing an example (good idea), it would be interesting to add two panels with vertical cross-sections of observed and modelled fog at the SIRTA site as well. Later, once you have developed your matching technique you could show the result on the same vertical cross-section to demonstrate the improvement visually as well.

**Response:** As the paper already contains a plot of the MRP values of simulated reflectivity compared to the nearest profile simulation and the observation, for a profile in which the MRP method is seen to work well, we only include a new figure comparing the observed and simulated reflectivity with the nearest model grid point for the case study presented in figure 1.

**Change:** Figure added for the first suggestion.

Added analysis of this figure line 255: Figure 2 shows the observed and simulated radar reflectivity profiles for the case on 04/11/2018 at two instances when there is fog recorded in the observations and predicted by the simulation. In both cases, the model overestimates the fog thickness, however, this overestimation is lower in the mature phase compared to the dissipation phase (30m vs 80m).

**Comment:** Line 237: " 12-hour window". Isn't that too big a window ? Atmospheric conditions can

change substantially, as well as radiative forcing over such a large window. As a result, you could get fog in the model developing from very different processes.

**Response:** To clarify, a 12-hour window means a maximum of 6 hours ahead or behind the observation. However, analysis of the formation/dissipation temporal errors presented in figure3 showed that 80% of temporal errors were less than ±3 hours. It means that when applying the MRP method, it is likely that model profiles within only a 3 hour window will be selected which should assure that the consistency in the fog processes.

**Comment:** Lines 247-248: it is very interesting to learn that the model tends to produce shorter fog episodes. Is that only when fog episodes are both observed and simulated or separately for all observed and all simulated? The reason why I ask is because you say that the model tends to produce a lot of cases that are not observed. What would that number be for the simulated when observed and simulated when not observed, any difference ?

**Response:** Yes this is for all events predicted by the model. It is an interesting point that the fog events that the model predicts when none is observed tend to be a lot shorter. This may also be partly because of the short forecast lead time (assimilated observations may cause the model to not predict fog in new cycle, thus causing to dissipate earlier than if 12- or 24-hour forecasts were used).

**Change:** Added: ' When all fog events observed and modelled are considered, modelled fog events tend to have a shorter duration, with an average fog time length of 4 hours 53 minutes (4H53M) compared to 6H03M for observed events, as many more short fog events were present in the model but not in the observations than vice versa. When only fog events present in the model and observation were compared, the mean duration of the modelled events was longer (6H44M for modelled events compared to 6H12M for observed events). '

**Comment:** Figure 3 and its interpretation. I think your statement that simulated fog top tends to be larger (and you should say higher not larger here) is not obvious from the figure. I'd rather stick to the bias value, maybe say that distribution of differences (is it model – obs by the way ?) is centred on zero with the std of 104m.

**Change:** Sentence added: 'Histogram of differences in the fog top height observed with the cloud radar and simulated by the AROME model (observation – simulation). '
and: 'larger' changed to 'higher'

Added: ' Figure 4 shows the distribution of fog top thickness differences where a positive thickness difference means an observed fog top higher than the simulated fog top. The figure shows that errors of up to 300m were found, and 44% of fog top height differences were greater than 100m. The mean height difference is -22.5m, and the standard deviation of fog top heights is 104m.'

**Comment:** Paragraph starting line 336. Those are very interesting results, as is the impact on reflectivity from Fig.5. However, maybe some combination of the possible values of these three parameters is unrealistic. Does the literature say anything about the co-variability of these parameters? For instance are high concentrations always associated with a narrower range of ν or the other parameter? That would help build a more realistic picture of the true variability of the reflectivity simulations if you can use such knowledge in your sensitivity analysis

**Response:** This was a point raised in some way by all three reviewers and so more research was done into previous studies looking at this.

Indeed, it seems that varying α and ν together with the quoted uncertainties probably gives an overestimation of the total uncertainty. The main effect that increasing both parameters have, is to narrow the size distribution spectra. Going back to the literature, most studies investigating these parameters fixed α at one or three and looked at the optimal values of ν (Geoffroy et al., 2010; Mazoyer, 2016). Where α is lower, ν is typically higher. For continental clouds in ICE-3, a value of α = 1 is used, so it was decided to recompute our results fixing this value here.

Regarding the varying of ν with N, the most honest answer here is that there is not enough in the literature to define rigorous bounds for the variance of one with the other in the context of fog (Geoffroy et al, 2010). It is a very interesting point, and one for which more research could be conducted with a new experimental dataset that will soon be available from the SOFOG3D campaign). We agree with the reviewer that it is important to make the reader aware of the current limitation that we hope could be surpassed when extra in-situ measurements are available.

Attempts have been made to find optimal values of the ν parameter during fog conditions. In the thesis of Marie Mazoyer, this was investigated in the context of optimising the gamma distribution shape to represent observed fog droplet size distributions. She looked at the droplet size distributions for 24 fog cases, and attempted to optimise the gamma distribution fit for the first, second and fifth moments. The plots in figure 3 below show that the standard deviation of errors between the idealised distribution and the observed distribution for the first and second moment are both reduced for increasing ν. However, for the fifth moment, an increase in the ν coefficient resulted in increased errors. For simulations of radar reflectivity, the sixth moment of the distribution would be important to model (as radar reflectivity is proportional to the $r^6$ where the Rayleigh approximation is valid), as well as the third moment of the distribution for making retrievals of LWC. It could therefore be interesting to repeat this study and optimise for the third and sixth moments. From this, a better estimation of the mean value and standard deviation of values of ν, for the specific use of using radar reflectivity to make retrievals of LWC, could be performed.

[Figure]

**Figure 3:** Errors between observed and predicted by assuming modified gamma distribution with α set to 1(left) and three (right) for the first, second and fifth moment of the distribution. Figure taken from Mazoyer (2016).

In a study which aimed to minimise first, second, fifth and sixth moments of the cloud droplet distribution errors, Geoffroy (2010) found that the optimal value of ν for α = 1 could be estimated from the LWC. This parametrisation gave optimised values of ν = 6.8-11.1 for typical values of

LWC inside a fog layer. This agreed well with the work of Miles (2000). In their study, a mean value of 8.7 was found for the ν parameter, with a standard deviation of 6.3. The uncertainty from simulated reflectivity resulting from the uncertainty of this parameter was therefore calculated with values one standard deviation above and below the mean values. T

Change: Section 4.1 rewritten. Lines 325-369:

In the pair of equations, $N(D)$ is the droplet number concentration where D is the droplet diameter. Coefficients a and b determine the mass-diameter relationship of the droplets, which, when applied to cloud droplets are well known due to their spherical nature, and are set at 524 and 3 respectively. α and ν are fixed coefficients referred to as the shape parameters and are set to 1 and 3 respectively in ICE-3 for cloud liquid droplet over land. $N_0$ is the total droplet concentration and is set to 300 in ICE-3 for liquid cloud over land. M is the liquid water content of the grid point in $kg.m^{-3}$.

The advantages of using this modified gamma distribution are that the shape and median diameter of the distribution are modified with the liquid water content and number concentration of the cloud. For example, when using the modified gamma distribution with a total concentration of $30cm^{-3}$, the median diameter will be greater than for a total concentration of $300cm^{-3}$, as illustrated in figure 5.

As all parameters of the modified gamma distribution except for the liquid water content are held constant in ICE-3, when radar simulations are made for cloud with a droplet size distribution which the parameters do not accurately describe, errors are likely to be made in the calculation of radar reflectivity. In order to assess this uncertainty, simulations were made on an AROME model profile in fog conditions, for which the size distribution parameters were perturbed. These perturbations would need to reflect potential variabilities seen in (continental liquid water) fog and low liquid cloud.

Microphysical observations have been investigated on fog events in previous works(Mazoyer et al., 2019; Podzimek et al, 1997)  which tend to show lower droplet concentrations than is prescribed for continental clouds in the ICE3 microphysical scheme (of $300cm^{-3)}$). From the works of Mazoyer (2016), which looked at median droplet concentrations for continental fog events, and Zhao (2019), which investigated the microphysics of continental boundary layer clouds, reasonable lower and upper bounds of the $N_0$ parameter of 30 and $300cm^{-3}$ were decided. Figure 5 shows the difference in cloud droplet distribution shapes when these two values are used.

As the α and ν parameters both affect the width of the size distribution (as may be seen in figure 5), it has been a common approach (Mazoyer, 2016;  Geoffroy et al. 2010) to fix α and to optimise the value of ν. The most frequently used values are  $α = 1$ (Liu et al, 2000) and $α = 3$  (Seifert et al, 2001). For this work, it was decided to use $α = 1$ which was shown by Mazoyer (2016)  to best represent fog droplet size distributions and also for consistency with the ICE-3 value.

From previous studies examining the value of ν where $α = 1$ (Geoffroy, 2010; Miles, 2000) it was decided that a range of $ν = 6.8$ to $11.1$ should be used. The modified gamma distribution with these values is shown in figure 5. Though there may be correlations between the LWC and the value of N and ν, a parameterisation for the values of ν and $N_0$ for fog in the context of cloud radar has yet to be performed. For this reason, the parameters ν and $N_0$ are treated as varying randomly for the purpose of investigating the uncertainty in simulated reflectivity.

**Comment:** Line 377: " occur around 90 min". Is that your visual inspection estimate or is it the value that your MRP process found?

**Change:** Thanks for the correction, this mistake was corrected in the new version 'event to occur 80 minutes...'

**Comment:.** Figure 6: I think you should mask out the profiles with precipitation in both observations  and model (using grey squares or something), as it is distracting from the main message of the figure.

**Change:** As suggested

**Comment:** Line 444 and Figure 10: One thing you should mention from this Figure, which is very positive, is that the distribution of errors is a lot narrower and around zero when MRP is used.

**Change:** As suggested on lne 498: 'The improvement in the standard deviation may be also seen in figure 11, in which the use of the MRP causes the distribution of reflectivity innovations to become narrower.'

---

## Author Comment (AC2)

**Manuscript Review: W-band Radar Observations for Fog Forecast Improvement: an Analysis of Model and Forward Operator Errors**

Thanks to all of the reviewers who kindly gave their time to analyse the article and returned some relevant comments to help clarify the research for future readers. We hope to respond to your queries below.

Reviewer 2

**Comment:** In the end, the authors have NOT yet assimilated radar reflectivity and demonstrated the improvement in fog forecast. Therefore, wording like "after selecting the best background profile, a good agreement was found between observations and simulations" in the abstract is really misleading. What the authors show is the agreement between observations and "selected background profiles", which is not surprising because the "best background profile" was selected using observations as a reference. If I somehow misunderstood the manuscript and if new simulations were indeed performed using the best background profiles, then this leads to an even bigger issue. By definition, the prior is NOT supposed to see the observations beforehand. Therefore, if new simulations were performed, they must be performed for a different case or time period, and I don't see any other cases different from those listed in Table 3. This kind of misleading statements can be found in Section 4.4 and Conclusions as well, which needs to be more precise.

**Response:** This is a good point- we have not yet shown that the background profiles are the best profiles for retrievals and therefore the wording in sentences such as these is too strong. To clarify, forecasts from the AROME model were not run again. The profiles were simply selected from the values of their simulated reflectivity. We expect that a better selection of background profiles will improve the data assimilation of radar reflectivity and therefore the fog forecast in future studies following this first part.

As you said, it is important in classical data assimilation techniques that the prior does not see the observations. However, for cases (such as we have shown for fog forecasts) where temporal and spatial errors can be significant, these techniques can also be ineffective (Ravela et al., 2007) and thus require an extra step before classical techniques can be applied. The method that we have shown in this paper therefore made use of a method to select a background profile which could account for temporal and spatial errors, without directly changing the background profile to suit the observations. In that sense, the selected background profile is still a forecast independent from the observation.

**Change:** Line 18 re-worded to 'After selecting the background profiles with the best agreement with the observations, the standard deviation of innovations (observations - simulations) was found to decrease significantly'

Line 432: 'Using the MRP selection, simulated reflectivity showed better agreement to observed reflectivities with the choice of a more appropriate background profile.'

Line 498: 'This study shows that, after removal of the largest background errors, the forward operator used in this study is able to replicate similar values of radar reflectivity from the background profiles, compared to the profiles observed during fog conditions'

Line 552: When a better agreement was found between the background profile and observation, the radar simulator was also found to be suitable to simulate the BASTA cloud radar reflectivity during fog conditions paving the way for larger model evaluations during fog events

**Comment:** Additionally, I think the use of "Innovations" is too strong and not accurate. It is an improvement, not an innovation.

**Response:** Here the term 'innovation' refers to observation minus simulation from the background profile $(y - H(x_b))$ values in the field of data assimilation.

**Change:** I've clarified the definition when the term is first used.

**Comment:** "Visibility measurements were averaged over 10-min period", meaning for both observations or simulations or both?

**Response:** This was only done for the observations, to account for noise – the model outputs were only available at a resolution of once per 10 minutes.

**Change:** Added: 'As model outputs were available with a temporal resolution of10 minutes, these were not averaged'

**Comment:** Descriptions in Section 3.2 are quite confusing to me, and I am not sure that readers are able to replicate results. What is "fog profile"? "Visibility measurements were averaged over 10-min period", meaning for both observations or simulations or both?

**Response:** It was used to describe a 10-minute time block where the model or observations were under fog conditions. I agree that this wording is not totally clear, so it has been re-written to clarify to the readers.

**Change:**
A comparison of observed fog to fog predicted in the model- for the time and grid point corresponding to the time and location of the observation- was carried out. Visibility measurements, taken from the DF-320 visibility sensor, were averaged over a 10 minute period, and where visibility values of lower than 1 km where observed, this was considered as a fog 'block'. The same threshold was used with visibility diagnosed from the model to define model fog 'blocks'. As model outputs were available with a temporal resolution of 10 minutes, these were not averaged. The accuracy of the model was then analysed by comparing each 10-minute block in the model against each block from the averaged visibility. Observations where rain was sensed with the rain gauge and simulations in which rain was present in the bottom layer were not considered as fog. The commonly used contingency table based on this comparison is shown in table 3 where GD indicates cases of good fog detection, FA cases of false alarm, ND cases of missed fog events by the model and CN correct negatives.

**Comment:** How might a choice of 28 km x 28 km domain relate to the sample size of 15248 in total in Table 4?

**Response:** As a 'fog block' in the model was diagnosed just from the grid point corresponding to the SIRTA observation site, the domain does not relate to the sample size mentioned in table 4. Instead, the sample size corresponds to one grid point every ten minutes for each day between 2[nd]

November until the 19[th] of February, with some blocks missing due to missing data, in which case no comparison was made.

**Comment:** The manuscript will read better if things are defined and clearly stated in a slightly different order. For example, how to define fog thickness in observations AND simulations? The term is introduced in 3.1 (page 7) but is not defined/explained until page 11. Even so, it is still unclear how exactly it is done and if it is the same for both observations and simulations... It would be nice to mention that earlier, so readers can connect Fig. 4 and the all exercises/results better.

**Response:** Thanks for the comment. We have tried to clarify what is meant by fog thickness in section 3.1. We did not want to discuss in depth how fog thickness is derived from the model in section 3.3, as in this section we do not discuss the comparison with observed fog thickness. However, when comparing model fog thickness/ fog top heights to observed fog thickness/fog top heights, in section 3.4, the way the fog top height is defined is important for the reader, and so we think it is more relevant to keep the explanation of fog top height prediction by the model in this section.

**Change:** Added in 3.3 (page 9) 'The fog thickness was diagnosed from simulated reflectivity values and is explained in more detail in section Section 3.4'

Added in line 290:  Fog thicknesses were derived from the radar observations during fog conditions. This was found from the height at which the radar reflectivity dropped below the larger of −45 dBZ or the sensitivity of the radar (whichever value  was greater) at that range gate.'

**Comment:** Another example is the information on parameter ranges on Page 15.

**Response:** In response to this comment and larger concerns of the other reviewers, the discussion of the parameters on page 15 has been significantly reviewed.

Change: Section 4.1 rewritten. Lines 325-369:

In the pair of equations, $N(D)$ is the droplet number concentration where $D$ is the droplet diameter. Coefficients $a$ and $b$ determine the mass-diameter relationship of the droplets, which, when applied to cloud droplets are well known due to their spherical nature, and are set at 524 and 3 respectively. $\alpha$ and $\nu$ are fixed coefficients referred to as the shape parameters and are set to 1 and 3 respectively in ICE-3 for cloud liquid droplet over land. $N_0$ is the total droplet concentration and is set to 300 in ICE-3 for liquid cloud over land. $M$ is the liquid water content of the grid point in $kg.m^{-3}$.

The advantages of using this modified gamma distribution are that the shape and median diameter of the distribution are modified with the liquid water content and number concentration of the cloud. For example, when using the modified gamma distribution with a total concentration of $30cm^{-3}$, the median diameter will be greater than for a total concentration of $300cm^{-3}$, as illustrated in figure 5.

As all parameters of the modified gamma distribution except for the liquid water content are held constant in ICE-3, when radar simulations are made for cloud with a droplet size distribution which the parameters do not accurately describe, errors are likely to be made in the calculation of radar reflectivity. In order to assess this uncertainty, simulations were made on an AROME model profile in fog conditions, for which the size distribution parameters were perturbed. These perturbations would need to reflect potential variabilities seen in (continental liquid water) fog and low liquid cloud.

Microphysical observations have been investigated on fog events in previous works(Mazoyer et al., 2019; Podzimek et al, 1997)  which tend to show lower droplet concentrations than is prescribed for continental clouds in the ICE3 microphysical scheme (of 300cm$^{-3)}$). From the works of Mazoyer (2016), which looked at median droplet concentrations for continental fog events, and Zhao (2019), which investigated the microphysics of continental boundary layer clouds, reasonable lower and upper bounds of the $N_0$ parameter of 30 and 300cm$^{-3}$ were decided. Figure 5 shows the difference in cloud droplet distribution shapes when these two values are used.

As the α and ν parameters both affect the width of the size distribution (as may be seen in figure 5), it has been a common approach (Mazoyer, 2016;  Geoffroy et al. 2010) to fix α and to optimise the value of ν. The most frequently used values are  α = 1 (Liu et al, 2000) and α = 3  (Seifert et al, 2001). For this work, it was decided to use α  = 1 which was shown by Mazoyer (2016)  to best represent fog droplet size distributions and also for consistency with the ICE-3 value.

From previous studies examining the value of ν where α  = 1 (Geoffroy, 2010; Miles, 2000) it was decided that a range of ν = 6.8 to 11.1 should be used. The modified gamma distribution with these values is shown in figure 5. Though there may be correlations between the LWC and the value of N and ν, a parameterisation for the values of ν and $N_0$ for fog in the context of cloud radar has yet to be performed. For this reason, the parameters ν and $N_0$ are treated as varying randomly for the purpose of investigating the uncertainty in simulated reflectivity.

**Comment:** Please explain why (c) only has 20 events? What happened to the other 11 events?

**Response:** Out of the 31 events observed, 21 could be matched to a modelled event.  Among the 11 missing events, 10 events could not be matched to a modelled event. For the events to be matched, fog must be present with a maximum of a 6 hour difference between the model and observation space (i.e. the dissipation time in one space can occur a maximum of 6 hours before the formation time in another). The last missing fog event was discarded due toformation time difference greater than six hours, hence it is not shown on the histogram.

**Change:** Added: 'Out of 31 fog events observed, 21 could be matched within the twelve hour window to a simulated event meaning that 10 observed events could not be matched to a modelled event'

**Comment:** If one wants to improve fog forecast, shouldn't we worry more about those 11 events? Can the authors comment if the newly selected background profiles will help improve the forecast for those 11 events?

**Response:** Indeed, it was for this reason that it was decided to use the MRP method to correct for spatial as well as temporal errors. However, if there is no fog forecast by the model anywhere in the extracted domain then the MRP method is not able to select background profiles which contain fog. The MRP method increases the number of background profiles containing fog for times when there is fog in the observations. A background profile containing fog was found for 73% of observation fog blocks using the MRP method, compared to 63% when the nearest grid point profile was used.

For the remaining 27% of cases where a background profile containing fog can still not be found, it may be necessary to use a climatological background profile for future retrievals.

**Comment:** Additionally, the caption is confusing. Do you mean "where the event "occurs/dissipates" later in the observations"? If statistics are derived using simulations minus observations, then it is best to be consistent throughout the manuscript (e.g., fig. 2 and fig.3).

**Response:** All statistics throughout the manuscript are for Observation -Simulation.

**Change:** Re-worded to 'fog formation time differences for matching events; fog dissipation time differences for  matching events (differences are positive where the fog forms/dissipates later in the observation).'

**Comment:** Do you mean fog thickness can be exchanged with fog top height, since the figure title is fog thickness, not top height?

**Response:** Yes, exactly. From what I understand, the two terms are often used interchangeably (e.g. Román-Cascón et al., 2015). as long as altitude is given above ground level.

**Change:** The first time the term is used (section 3.1) I have indicated that the two terms can be used interchangeably, but have replaced fog top height with fog thickness where the term directly references a figure with this term.

---

## Author Comment (AC3)

**Manuscript Review: W-band Radar Observations for Fog Forecast Improvement: an Analysis of Model and Forward Operator Errors**

Thanks to all of the reviewers who kindly gave their time to analyse the article and returned some relevant comments to help clarify the research for future readers. We hope to respond to your queries below.

**Reviewer 3**

**Comment:** I find the simulations with the forward operator quite confused. First a Gamma modified PSD is introduced; this has 4 free parameters (the authors do not mention any correlation between parameters). They then introduce other 2 parameters C and X (I do not know really why?). For some reason then they study the variability of a profile with alpha, nu and N\_0 but they forget completely Lambda (i.e. the characteristic fog size). Why?

**Response:** The idea here was just to clarify all parameters in the modified gamma distribution as it is used in ICE-3. Indeed, the formulation of N with C and x is unnecessary here, as for liquid cloud droplets x = 0. For lambda, it would also be possible to modify this parameterisation to investigate the uncertainty in this parameter, however as it is already a function of the mass of droplets,  $\alpha$  and v, it had been indirectly modified through changes in the v parameter already.

**Change:** Several paragraphs have been added in section 4.1 (detailed below) to clarify the work done here and to summarise previous works.

Equation (7) was removed and equation (6) has been adapted to make sense without mention of these parameters.

**Comment:** Fig.4: all units in the y-axis are wrong. Not sure how useful is Fig.4, particularly the bottom panel. If N\_0 changes then there is just an amplification (not sure the figure is actually right, it looks like the maximum of the blue line is different from the orange one). Similarly simulating reflectivities changing N\_0 is trivial and should not be plotted (Fig 5, right panel), doubling N\_0 just add 3 dB.

**Response:** It seems that some additional explanation was needed before introducing the figures for a better understanding. Indeed, the maxima of this modified gamma distribution will change when the concentration number is changed in order to keep the liquid water content the same, so this figure is correct.

The increase in reflectivity would indeed be 3dB per doubling of the droplets *if the original distribution was doubled*. However, this is not what this work shows. In fact, it shows that increasing the number concentration *reduces* the values of reflectivity, because there are more small droplets but fewer large droplets. This is because the LWC is conserved in the modified gamma distribution. In our opinion, (old figure 5b, new figure 6b) is necessary to compare to (old figure 5a, new Figure 6a) to show the magnitude of the changes in reflectivity when the two parameters are changed.

**Change:** Thanks for noting the axis unit error- figures have been remade correcting for the errors in the scale and units on the y-axis. These figures were also remade using a logarithmic scale so that the changes in the distribution shape and the absolute values can be clearly seen.

Inserted paragraph before figures of distributions: 'The advantages of using the modified gamma distribution are that the shape and median diameter of the distribution are modified with the liquid water content of the cloud. For example, when using the modified gamma distribution with a total concentration of 30 cm-3, the median diameter will be greater than for a total concentration of 300 cm-3, as illustrated in figure 5. This is because the same amount of water must be divided among fewer droplets.'

**Comment:** On the other hand the change of alpha nu and Lambda should be better investigated accounting for the possible relationship between the different parameters (It is not enough to change only one parameter at a time).

**Response:** This was a point raised in some way by all three reviewers and so more research was done into previous studies looking at this.

Indeed, it seems that varying  $\alpha$  and  $\nu$  together with the quoted uncertainties probably gives an overestimation of the total uncertainty. The main effect that increasing both parameters have, is to narrow the size distribution spectra. Going back to the literature, most studies investigating these parameters fixed  $\alpha$  at one or three and looked at the optimal values of  $\nu$  (Geoffroy et al., 2010; Mazoyer, 2016). Where  $\alpha$  is lower,  $\nu$  is typically higher. For continental clouds in ICE-3, a value of  $\alpha = 1$  is used, so it was decided to recompute our results fixing this value here.

Regarding the varying of v with N, the most honest answer here is that there is not enough in the literature to define rigorous bounds for the variance of one with the other in the context of fog (Geoffroy et al, 2010). It is a very interesting point, and one for which more research could be conducted with a new experimental dataset that will soon be available from the SOFOG3D campaign). We agree with the reviewer that it is important to make the reader aware of the current limitation that we hope could be surpassed when extra in-situ measurements are available.

Attempts have been made to find optimal values of the v parameter during fog conditions. In the thesis of Marie Mazoyer, this was investigated in the context of optimising the gamma distribution shape to represent observed fog droplet size distributions. She looked at the droplet size distributions for 24 fog cases, and attempted to optimise the gamma distribution fit for the first, second and fifth moments. The plots in figure 3 below show that the standard deviation of errors between the idealised distribution and the observed distribution for the first and second moment are both reduced for increasing v. However, for the fifth moment, an increase in the v coefficient resulted in increased errors. For simulations of radar reflectivity, the sixth moment of the distribution would be important to model (as radar reflectivity is proportional to the r6 where the Rayleigh approximation is valid), as well as the third moment of the distribution for making retrievals of LWC. It could therefore be interesting to repeat this study and optimise for the third and sixth moments. From this, a better estimation of the mean value and standard deviation of values of v, for the specific use of using radar reflectivity to make retrievals of LWC, could be performed.

**Figure 3:** Errors between observed and predicted by assuming modified gamma distribution with  $\alpha$  set to 1(left) and three (right) for the first, second and fifth moment of the distribution. Figure taken from Mazoyer (2016).

In a study which aimed to minimise first, second, fifth and sixth moments of the cloud droplet distribution errors, Geoffroy (2010) found that the optimal value of v for  $\alpha = 1$  could be estimated from the LWC. This parametrisation gave optimised values of v = 6.8-11.1 for typical values of LWC inside a fog layer. This agreed well with the work of Miles (2000). In their study, a mean value of 8.7 was found for the v parameter, with a standard deviation of 6.3. The uncertainty from simulated reflectivity resulting from the uncertainty of this parameter was therefore calculated with values one standard deviation above and below the mean values. T

Change: Section 4.1 rewritten. Lines 325-369:

In the pair of equations, N(D) is the droplet number concentration where D is the droplet diameter. Coefficients a and b determine the mass-diameter relationship of the droplets, which, when applied to cloud droplets are well known due to their spherical nature, and are set at 524 and 3 respectively.  $\alpha$  and v are fixed coefficients referred to as the shape parameters and are set to 1 and 3 respectively in ICE-3 for cloud liquid droplet over land. N0 is the total droplet concentration and is set to 300 in ICE-3 for liquid cloud over land. M is the liquid water content of the grid point in kg.m-3.

The advantages of using this modified gamma distribution are that the shape and median diameter of the distribution are modified with the liquid water content and number concentration of the cloud. For example, when using the modified gamma distribution with a total concentration of 30cm-3, the median diameter will be greater than for a total concentration of 300cm-3, as illustrated in figure 5.

As all parameters of the modified gamma distribution except for the liquid water content are held constant in ICE-3, when radar simulations are made for cloud with a droplet size distribution which the parameters do not accurately describe, errors are likely to be made in the calculation of radar reflectivity. In order to assess this uncertainty, simulations were made on an AROME model profile in fog conditions, for which the size distribution parameters were perturbed. These perturbations would need to reflect potential variabilities seen in (continental liquid water) fog and low liquid cloud.

Microphysical observations have been investigated on fog events in previous works(Mazoyer et al., 2019; Podzimek et al, 1997) which tend to show lower droplet concentrations than is prescribed for continental clouds in the ICE3 microphysical scheme (of 300cm-3)). From the works of Mazoyer (2016), which looked at median droplet concentrations for continental fog events, and Zhao (2019), which investigated the microphysics of continental boundary layer clouds, reasonable lower and

upper bounds of the  $N_0$  parameter of 30 and 300cm-3 were decided. Figure 5 shows the difference in cloud droplet distribution shapes when these two values are used.

As the  $\alpha$  and  $\nu$  parameters both affect the width of the size distribution (as may be seen in figure 5), it has been a common approach (Mazoyer, 2016; Geoffroy et al. 2010) to fix  $\alpha$  and to optimise the value of  $\nu$ . The most frequently used values are  $\alpha = 1$  (Liu et al, 2000) and  $\alpha = 3$  (Seifert et al, 2001). For this work, it was decided to use  $\alpha = 1$  which was shown by Mazoyer (2016) to best represent fog droplet size distributions and also for consistency with the ICE-3 value.

From previous studies examining the value of v where  $\alpha = 1$  (Geoffroy, 2010; Miles, 2000) it was decided that a range of v = 6.8 to 11.1 should be used. The modified gamma distribution with these values is shown in figure 5. Though there may be correlations between the LWC and the value of N and v, a parameterisation for the values of v and N0 for fog in the context of cloud radar has yet to be performed. For this reason, the parameters v and N0 are treated as varying randomly for the purpose of investigating the uncertainty in simulated reflectivity.

**Comment:** Line 405-410: I am not convinced that some of the big differences we see in Fig.6 can be attributed to non sphericity. Where is the freezing level in this scene? Also instead of ``isotropic particles" use ``spherical particles".

**Response:** I think the placement of figures in the article made this a little confusing- indeed it was not my intention to suggest the differences in figure 6 arise from non sphericity, it is just that the figure which is commented upon in the previous section appears with the text where the sphericity is discussed. However, I will say that it was verified that the fog in this case was below the freezing level in both observations and simulations.

Changes: 'isotropic' changed to 'spherical'

Figure has also been masked where large differences between simulation and observation occur due to precipitation.

**Comment:** Fig8: not sure about the cluster of points above 500 m. Is that fog? If so why you are cutting the plots at 1km?

**Response:** The clusters at around 700-800m are indeed clouds. The plots were cut at 1km as in winter above 1km we more commonly see ice in clouds and it was not the objective of this paper to examine the ice clouds. However, even if this work focuses on fog, low clouds have a significant impact on the fog life cycle with potential stratus lowering. It is thus important to validate our methodology also for the low liquid clouds.

**Comment:** Tab1: Range for HATPRO (0 to 10 km) ==> it does not make any sense to specify a range for a radiometer

Change: Agreed. This has been removed.

**Comment:** Fig1, caption: I do not see 11:00 UTC but 10:20 UTC in the plots.

**Change:** Thanks for the correction. Caption changed as advised.